# Generalised Mutual Information for Discriminative Clustering

**Louis Ohl**
Université Côte d'Azur
Inria, CNRS
I3S, Maasai team
CHU de Québec Research Center
Laval University
louis.ohl@inria.fr

**Pierre-Alexandre Mattei**
Université Côte d'Azur
Inria, CNRS
LJAD, Maasai team
pierre-alexandre.mattei@inria.fr

**Charles Bouveyron**
Université Côte d'Azur
Inria, CNRS
LJAD, Maasai team

**Warith Harchaoui**
Jellysmack
AI Labs
Research and Development

**Mickael Leclerq**
CHU de Québec Research Center
Laval University

**Arnaud Droit**
CHU de Québec Research Center
Laval University

**Frederic Precioso**
Université Côte d'Azur
Inria, CNRS
I3S, Maasai team

## Abstract

In the last decade, recent successes in deep clustering majorly involved the mutual information (MI) as an unsupervised objective for training neural networks with increasing regularisations. While the quality of the regularisations have been largely discussed for improvements, little attention has been dedicated to the relevance of MI as a clustering objective. In this paper, we first highlight how the maximisation of MI does not lead to satisfying clusters. We identified the Kullback-Leibler divergence as the main reason of this behaviour. Hence, we generalise the mutual information by changing its core distance, introducing the generalised mutual information (GEMINI): a set of metrics for unsupervised neural network training. Unlike MI, some GEMINIs do not require regularisations when training. Some of these metrics are geometry-aware thanks to distances or kernels in the data space. Finally, we highlight that GEMINIs can automatically select a relevant number of clusters, a property that has been little studied in deep clustering context where the number of clusters is a priori unknown.

## 1 Introduction

Clustering is a fundamental learning task which consists in separating data samples into several categories, each named cluster. This task hinges on two main questions concerning the assessment of correct clustering and the actual number of clusters that may be contained within the data distribution. However, this problem is ill-posed since a cluster lacks formal definitions which makes it a hard problem (Kleinberg, 2003).

Model-based algorithms make assumptions about the true distribution of the data as a result of some latent distribution of clusters (Bouveyron et al., 2019). These techniques are able to find the most

36th Conference on Neural Information Processing Systems (NeurIPS 2022).

likely cluster assignment to data points. These models are usually generative, exhibiting an explicit assumption of the prior knowledge on the data.

Early deep models to perform clustering first relied on autoencoders, based on the belief that an encoding space holds satisfactory properties (Xie et al., 2016; Ghasedi Dizaji et al., 2017; Ji et al., 2019). However, the drawback of these architectures is that they do not guarantee that data samples which should meaningfully be far apart remain so in the feature space. Early models that dropped decoders notably used the mutual information (MI) (Krause et al., 2010; Hu et al., 2017) as an objective to maximise. The MI can be written in two ways, either as measure of dependency between two variables $x$ and $y$, e.g. data distribution $p(x)$ and cluster assignment $p(y)$:

$$\mathcal{I}(x;y) = D_{\mathrm{KL}}(p(x,y)||p(x)p(y)), \tag{1}$$

or as an expected distance between implied distributions and the overall data:

$$\mathcal{I}(x;y) = \mathbb{E}_{p(y)}[D_{\mathrm{KL}}(p(x|y)||p(x))], \tag{2}$$

with $D_{\mathrm{KL}}$ being the Kullback-Leibler (KL) divergence. Related works often relied on the notion of MI as a measure of coherence between cluster assignments and data distribution (Hjelm et al., 2019). Regularisation techniques were employed to leverage the potential of MI, mostly by specifying model invariances, for example with data augmentation (Ji et al., 2019).

The maximisation of MI thus gave way to contrastive learning objectives which aim at learning stable representations of data through such invariance specifications (Chen et al., 2020; Caron et al., 2020). The contrastive loss maximises the similarity between the features of a sample and its augmentation, while decreasing the similarity with any other sample. Clustering methods also benefited from recent successful deep architectures (Li et al., 2021; Tao et al., 2021; Huang et al., 2020) by encompassing regularisations in the architecture. These methods correspond to discriminative clustering where we seek to directly infer cluster given the data distribution. Initial methods also focused on alternate schemes, for example with curriculum learning (Chang et al., 2017) to iteratively select relevant data samples for training: for example by alternating K-means cluster assignment with supervised learning using the inferred labels (Caron et al., 2018), or by proceeding to multiple distinct training steps (Van Gansbeke et al., 2020; Dang et al., 2021; Park et al., 2021).

However, most of the methods above rarely discuss their robustness when the number of clusters to find is different from the amount of preexisting known classes. While previous work was essentially motivated by considering MI as a dependence measure, we explore in this paper the alternative definition of the MI as the expected distance between data distribution implied by the clusters and the entire data. We extend it to incorporate cluster-wise comparisons of implied distributions, and question the choice of the KL divergence with other possible statistical distances.

Throughout the introduction of the generalised mutual information (GEMINI), the contributions of this paper are:

- A demonstration of how the maxima of MI are not sufficient criteria for clustering. This extends the contribution of (Tschannen et al., 2020) to the discrete case.

- The introduction of a set of metrics called GEMINIs involving different distances between distributions which can incorporate prior knowledge on the geometry of the data. Some of these metrics do not require regularisations.

- A highlight of the implicit selection of clusters from GEMINIs which allows to select a relevant number of cluster during training.

## 2 Is mutual information a good clustering objective?

We consider in this section a dataset consisting in $N$ unlabelled samples $\mathcal{D} = \{\boldsymbol{x}_i\}_{i=1}^{N}$. We distinguish two major use cases of the mutual information: one where we measure the dependence between two continuous variables, as is the case in representation learning, and one where the random variable is discrete. In representation learning, the goal is to construct a continuous representation $\boldsymbol{z}$ extracted from the data $\boldsymbol{x}$ using a learnable distribution of parameters $\theta$. In clustering, samples $\boldsymbol{x}$ are assigned to the discrete variable $y$ through another learnable distribution.

## 2.1 Representation learning

Representation learning consists in finding high-level features $z_i$ extracted from the data $x_i$ in order to perform a *downstream task*, e.g. clustering or classification. MI between $x$ and $z$ is a common choice for learning features(Hjelm et al., 2019). However, estimating correctly MI between two random variables in continuous domains is often intractable when $p(x|z)$ or $p(z|x)$ is unknown, thus lower bounds are preferred, e.g. variational estimators such as MINE (Belghazi et al., 2018), $\mathcal{I}_{\text{NCE}}$(Van den Oord et al., 2018). Another common choice of loss function to train features are contrastive losses such as NT-XENT (Chen et al., 2020) where the similarity between the features $z_i$ from data $x_i$ is maximised with the features $\tilde{z}$ from a data-augmented $\tilde{x}_i$ against any other features $z_j$. Recently, Do et al. (2021) achieved excellent performances in single-stage methods by highlighting the link between the $\mathcal{I}_{\text{NCE}}$ estimator (Van den Oord et al., 2018) and contrastive learning losses. Representation learning therefore comes at the cost of a complex lower bound estimator on MI, which often requires data augmentation. Moreover, it was noticed that the MI is hardly predictive of downstream tasks (Tschannen et al., 2020) when the variable $y$ is continuous, i.e. a high value of MI does not clarify whether the discovered representations are insightful with regards to the target of the downstream task.

## 2.2 Discriminative clustering

The MI has been first used as an objective for learning discriminative clustering models (Bridle et al., 1992). Associated architectures went from simple logistic regression (Krause et al., 2010) to deeper architectures (Hu et al., 2017; Ji et al., 2019). Beyond architecture improvement, the MI maximisation was also carried with several regularisations. These regularisations include penalty terms such as weight decay (Krause et al., 2010) or Virtual Adversarial Training (VAT, Hu et al., 2017; Miyato et al., 2018b). Data augmentation was further used to provide invariances in clustering, as well as specific architecture designs like auxiliary clustering heads (Ji et al., 2019). Rewriting the MI in terms of entropies:

$$\mathcal{I}(x; y) = \mathcal{H}(y) - \mathcal{H}(y|x) \tag{3}$$

highlights a requirement for balanced clusters, through the cluster entropy term $\mathcal{H}(y)$. Indeed, a uniform distribution maximises the entropy. This hints that an unregularised discrete mutual information for clustering can possibly produce uniformly distributed clusters among samples, regardless of how close they could be. We highlight this claim in section 2.3. As an example of regularisation impact: maximising the MI with $\ell_2$ constraint can be equivalent to a soft and regularised K-Means in a feature space (Jabi et al., 2019). In clustering, the number of clusters to find is usually not known in advance. Therefore, an interesting clustering algorithm should be able to find a relevant number of clusters, i.e. perform model selection. However, model selection for parametric deep clustering models is expensive (Ronen et al., 2022). Cluster selection through MI maximisation has been little studied in related works, since experiments usually tasked models to find the (supervised) classes of datasets. Furthermore, the literature diverged towards deep learning methods focusing mainly on images, yet rarely on other type of data such as tabular data (Min et al., 2018).

## 2.3 Maximising the MI can lead to bad decision boundaries

Maximising the MI directly can be a poor objective: a high MI value is not necessarily predictive of the quality of the features regarding downstream tasks (Tschannen et al., 2020) when $y$ is continuous. We support a similar argument for the case where the data $x$ is a continuous random variable and the cluster assignment $y$ a categorical variable. Indeed, the MI can be maximised by setting appropriately a sharp decision boundary which partitions evenly the data. This reasoning can be seen in the entropy-based formulation of the MI (Eq. 3): any sharp decision boundary minimises the negative conditional entropy, while ensuring balanced clusters maximises the entropy of cluster proportions. Consider for example Figure 1, where a mixture of Gaussian distributions with equal variances is separated by a sharp decision boundary. We highlight that both models will have the same mutual information on condition that the misplaced decision boundary of Figure 1b splits evenly the dataset (see Appendix A).

Globally, MI misses the idea in clustering that any two points close to one another may be in the same cluster according to some chosen metric. Hence regularisations are required to ensure this constraint. A sketch of these insights was mentionned by Bridle et al. (1992); Corduneanu and Jaakkola (2002).

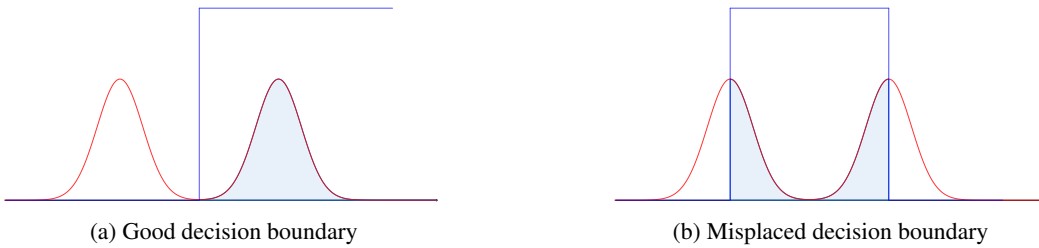

| (a) Good decision boundary | (b) Misplaced decision boundary |

Figure 1: Example of maximised MI for a Gaussian mixture $p(x) \sim \frac{1}{2}\mathcal{N}(\mu_0, \sigma^2) + \frac{1}{2}\mathcal{N}(\mu_1, \sigma^2)$. It is clear that Figure 1a presents the best decision boundary and posterior between the two Gaussian distributions. Yet, as $p(x|y)$ converges to a Dirac distribution, the MI difference converges to 0.

## 3 Extending the mutual information to the generalised mutual information

Given the identified limitations of MI, we now describe the discriminative clustering framework based on our perception of the mutual information. We then detail the different statistical distances we can use to extend MI to the generalised mutual information (GEMINI).

### 3.1 The discriminative clustering framework for GEMINIs

We change our view on the mutual information by seeing it as a discriminative clustering objective that aims at separating the data distribution given cluster assignments $p(\boldsymbol{x}|y)$ from the data distribution $p(\boldsymbol{x})$ according to the KL divergence:

$$\mathcal{I}(\boldsymbol{x}; y) = \mathbb{E}_{y \sim p(y)} \left[ D_{\mathrm{KL}}(p(\boldsymbol{x}|y) \| p(\boldsymbol{x})) \right]. \tag{4}$$

To highlight the discriminative clustering design, we explicitly remove our hypotheses on the data distribution by writing $p_{\mathrm{data}}(\boldsymbol{x})$. The only part of the model that we design is a conditional distribution $p_\theta(y|\boldsymbol{x})$ that assigns a cluster $y$ to a sample $\boldsymbol{x}$ using the parameters $\theta$ (Minka, 2005). This conditional distribution can typically be a neural network of adequate design regarding the data, e.g. a CNN, or a simple categorical distribution. Consequently, the cluster proportions are controlled by $\theta$ because $p_\theta(y) = \mathbb{E}[p_\theta(y|\boldsymbol{x})]$ and so is the conditional distribution $p_\theta(\boldsymbol{x}|y)$ even though intractable. This questions how Eq. (4) can be computed. Fortunately, well-known properties of MI can invert the distributions on which the KL divergence is computed (Bridle et al., 1992; Krause et al., 2010) via Bayes' theorem:

$$\mathcal{I}(\boldsymbol{x}; y) = \mathbb{E}_{\boldsymbol{x} \sim p_{\mathrm{data}}(\boldsymbol{x})} \left[ D_{\mathrm{KL}}(p_\theta(y|\boldsymbol{x}) \| p_\theta(y)) \right], \tag{5}$$

which is possible to estimate. Since we highlighted earlier that the KL divergence in the MI can lead to inappropriate decision boundaries, we are interested in replacing it by other distances or divergences. However, changing it in Eq. (5) would focus on the separation of cluster assignments from the cluster proportions which may be irrelevant to the data distribution. We rather alter Eq. (4) to clearly show that we separate data distributions given clusters from the entire data distribution because it allows us to take into account the data space geometry.

### 3.2 The GEMINI

The goal of the GEMINI is to separate data distributions according to an arbitrary distance $D$, i.e. changing the KL divergence for another divergence or distance in the MI. Moreover, we question the evaluation of the distance between the distribution of the data given a cluster assumption $p_\theta(\boldsymbol{x}|y)$ and the entire data distribution $p_{\mathrm{data}}(\boldsymbol{x})$. We argue that it is intuitive in clustering to compare the distribution of one cluster against the distribution of *another cluster* rather than the data distribution. This raises the definition of two GEMINIs, one named *one-vs-all* (OvA):

$$\mathcal{I}_D^{\mathrm{OvA}}(\boldsymbol{x}; y) = \mathbb{E}_{y \sim p_\theta(y)} \left[ D(p_\theta(\boldsymbol{x}|y) \| p_{\mathrm{data}}(\boldsymbol{x})) \right], \tag{6}$$

which compares the cluster distributions to the data distribution, and the *one-vs-one* (OvO) in which we independently draw cluster assignments $y_a$ and $y_b$ (see App. B for an OvO justification):

$$\mathcal{I}_D^{\mathrm{OvO}}(\boldsymbol{x}; y) = \mathbb{E}_{y_a, y_b \sim p_\theta(y)} \left[ D(p_\theta(\boldsymbol{x}|y_a) \| p_\theta(\boldsymbol{x}|y_b)) \right], \tag{7}$$

Table 1: Definition of the GEMINI for $f$-divergences, MMD and the Wasserstein distance. We directly write here the equation that can be optimised to train a discriminative model $p_\theta(y|\boldsymbol{x})$ via stochastic gradient descent since they are expectations over the data.

| Name | Equation |
|---|---|
| KL OvA/MI | $\mathbb{E}_{p_{\text{data}}(\boldsymbol{x})}\left[D_{\text{KL}}(p_\theta(y|\boldsymbol{x})\|p_\theta(y))\right]$ |
| KL OvO | $\mathbb{E}_{p_{\text{data}}(\boldsymbol{x})}[D_{\text{KL}}(p_\theta(y|\boldsymbol{x})\|p_\theta(y))) + D_{\text{KL}}(p_\theta(y)\|p_\theta(y|\boldsymbol{x})))]$ |
| Squared Hellinger OvA | $1 - \mathbb{E}_{p_{\text{data}}(\boldsymbol{x})}[\mathbb{E}_{p_\theta(y)}[\sqrt{\frac{p_\theta(y|\boldsymbol{x})}{p_\theta(y)}}]]$ |
| Squared Hellinger OvO | $\mathbb{E}_{p_{\text{data}}(\boldsymbol{x})}[\mathbb{V}_{p_\theta(y)}[\sqrt{\frac{p_\theta(y|\boldsymbol{x})}{p_\theta(y)}}]]$ |
| TV OvA | $\mathbb{E}_{p_{\text{data}}(\boldsymbol{x})}[D_{\text{TV}}(p_\theta(y|\boldsymbol{x})\|p_\theta(y))]$ |
| TV OvO | $\frac{1}{2}\mathbb{E}_{p_{\text{data}}(\boldsymbol{x})}[\mathbb{E}_{y_a,y_b\sim p_\theta(y)}[|\frac{p_\theta(y_a|\boldsymbol{x})}{p_\theta(y_a)} - \frac{p_\theta(y_b|\boldsymbol{x})}{p_\theta(y_b)}|]]$ |
| MMD OvA | $\mathbb{E}_{p_\theta(y)}\left[\mathbb{E}_{\boldsymbol{x}_a,\boldsymbol{x}_b\sim p_{\text{data}}(\boldsymbol{x})}\left[k(\boldsymbol{x}_a,\boldsymbol{x}_b)\left(\frac{p_\theta(y|\boldsymbol{x}_a)p_\theta(y|\boldsymbol{x}_b)}{p_\theta(y)^2} + 1 - 2\frac{p_\theta(y|\boldsymbol{x}_a)}{p_\theta(y)}\right)\right]^{\frac{1}{2}}\right]$ |
| MMD OvO | $\mathbb{E}_{y_a,y_b\sim p_\theta(y)}\left[\mathbb{E}_{\boldsymbol{x}_a,\boldsymbol{x}_b\sim p_{\text{data}}(\boldsymbol{x})}\left[k(\boldsymbol{x}_a,\boldsymbol{x}_b)\left(\frac{p_\theta(y_a|\boldsymbol{x}_a)p_\theta(y_a|\boldsymbol{x}_b)}{p_\theta(y_a)^2}\right.\right.\right.$ $\left.\left.\left. + \frac{p_\theta(y_b|\boldsymbol{x}_a)p_\theta(y_b|\boldsymbol{x}_b)}{p_\theta(y_b)^2} - 2\frac{p_\theta(y_a|\boldsymbol{x}_a)p_\theta(y_b|\boldsymbol{x}_b)}{p_\theta(y_a)p_\theta(y_b)}\right)\right]^{\frac{1}{2}}\right]$ |
| Wasserstein OvA | $\mathbb{E}_{p_\theta(y)}\left[\mathcal{W}_c\left(\sum_{i=1}^N m_i^y \delta_{\boldsymbol{x}_i}, \sum_{i=1}^N \frac{1}{N}\delta_{\boldsymbol{x}_i}\right)\right]$ |
| Wasserstein OvO | $\mathbb{E}_{y_a,y_b\sim p_\theta(y)}\left[\mathcal{W}_c\left(\sum_{i=1}^N m_i^{y_a}\delta_{\boldsymbol{x}_i}, \sum_{i=1}^N m_i^{y_b}\delta_{\boldsymbol{x}_i}\right)\right]$ |

There exists other distances than the KL to measure how far two distributions $p$ and $q$ are one from the other. We can make a clear distinction between two types of distances, Csiszar's $f$-divergences (Csiszár, 1967) and integral probability metrics (IPM) (Sriperumbudur et al., 2009). However, unlike $f$-divergences, IPM-derived distances like the Wasserstein distance or the maximum mean discrepancy (MMD)(Gneiting and Raftery, 2007; Gretton et al., 2012) bring knowledge about the data throughout either a distance $c$ or a kernel $\kappa$: these distances are geometry-aware.

$f$**-divergence GEMINIs:** These divergences involve a convex function $f : \mathbb{R}^+ \to \mathbb{R}$ such that $f(1) = 0$. This function is applied to evaluate the ratio between two distributions $p$ and $q$, as in Eq. (8):

$$D_{\text{f-div}}(p,q) = \mathbb{E}_{\boldsymbol{z}\sim q(\boldsymbol{z})}\left[f\left(\frac{p(\boldsymbol{z})}{q(\boldsymbol{z})}\right)\right]. \tag{8}$$

We will focus on three $f$-divergences: the KL divergence, the total variation (TV) distance and the squared Hellinger distance. While the KL divergence is the usual divergence for the MI, the TV and the squared Hellinger distance present different advantages among $f$-divergences. First of all, both of them are bounded between 0 and 1. It is consequently easy to check when any GEMINI using those is maximised contrarily to the MI that is bounded by the minimum of the entropies of $\boldsymbol{x}$ and $y$ (Gray and Shields, 1977). When used as distance between data conditional distribution $p_\theta(\boldsymbol{x}|y)$ and data distribution $p_{\text{data}}(\boldsymbol{x})$, we can apply Bayes' theorem in order to get an estimable equation to maximise, which only involves cluster assignment $p_\theta(y|\boldsymbol{x})$ and marginals $p_\theta(y)$ (see Table 1).

**MMD GEMINIs:** The MMD corresponds to the distance between the respective expected embedding of samples from distribution $p$ and distribution $q$ in a reproducing kernel hilbert space (RKHS) $\mathcal{H}$:

$$\text{MMD}(p\|q) = \|\mathbb{E}_{\boldsymbol{z}\sim p(\boldsymbol{z})}[\varphi(\boldsymbol{z})] - \mathbb{E}_{\boldsymbol{z}\sim q(\boldsymbol{z})}[\varphi(\boldsymbol{z})]\|_{\mathcal{H}}, \tag{9}$$

where $\varphi$ is the RKHS embedding. To compute this distance we can use the kernel trick (Gretton et al., 2012) by involving the kernel function $\kappa(\boldsymbol{a},\boldsymbol{b}) = \langle\varphi(\boldsymbol{a}),\varphi(\boldsymbol{b})\rangle$. We use Bayes' theorem to uncover a version of the MMD that can be estimated through Monte Carlo using only the predictions $p_\theta(y|\boldsymbol{x})$ (see Table 1).

**Wasserstein GEMINIs:** This distance is an optimal transport distance, defined as:

$$\mathcal{W}_c(p, q) = \left( \inf_{\gamma \in \Gamma(p,q)} \mathbb{E}_{\boldsymbol{x}, \boldsymbol{z} \sim \gamma(\boldsymbol{x}, \boldsymbol{z})} \left[ c(\boldsymbol{x}, \boldsymbol{z}) \right] \right), \tag{10}$$

where $\Gamma(p, q)$ is the set of all couplings between $p$ and $q$ and $c$ a distance function in $\mathcal{X}$. Computing the Wasserstein distance between two distributions $p_\theta(\boldsymbol{x}|y = k_1)$ and $p_\theta(\boldsymbol{x}|y = k_2)$ is difficult in our discriminative context because we only have access to a finite set of samples $N$. To achieve the Wasserstein-GEMINI, we instead use approximations of the distributions with weighted sums of Diracs:

$$p_\theta(\boldsymbol{x}|y = k) \approx \sum_{i=1}^{N} m_i^k \delta_{\boldsymbol{x}_i} = p_N^k, \quad \text{with} \quad m_i^k = \frac{p_\theta(y = k|\boldsymbol{x}_i)}{\sum_{j=1}^{N} p_\theta(y = k|\boldsymbol{x}_j)}, \tag{11}$$

where $\delta_{\boldsymbol{x}_i}$ is a Dirac located on sample location $\boldsymbol{x}_i \in \mathcal{X}$. The Wasserstein-OvA and -OvO applied to Dirac sums are compatible with the `emd2` function of the Python optimal transport package (Flamary et al., 2021) which gracefully supports automatic differentiation (see Appendix D.3 for convergence to the expectation). All GEMINIs are summarised in Table 1, (see Appendix D for derivations).

# 4 Experiments

For all experiments below, we report the adjusted rand index (ARI) (Hubert and Arabie, 1985), a common metric in clustering. This metric is external as it requires labels for evaluation. It ranges from 0, when labels are independent from cluster assignments, to 1, when labels are equivalent to cluster assignments up to permutations. An ARI close to 0 is equivalent to the best accuracy when voting constantly for the majority class, e.g. 10% on a balanced 10-class dataset. Regarding the MMD- and Wasserstein-GEMINIs, we used by default a linear kernel and the Euclidean distance unless specified otherwise. All discriminative models are trained using the Adam optimiser (Kingma and Ba, 2014). We estimate a total of 450 hours of GPU consumption. (See Appendix I for the details of Python packages for experiments and Appendix F for further experiments regarding model selection). The code is available at `https://github.com/oshillou/GEMINI`

## 4.1 When the MI fails because of the modelling

We first took the most simple discriminative clustering model, where each cluster assignment according to the input datum follows a categorical distribution:

$$y|\boldsymbol{x} = \boldsymbol{x}_i \sim \text{Cat}(\theta_i^1, \theta_i^2, \cdots, \theta_i^K).$$

We generated $N = 100$ samples from a simple mixture of $K = 3$ Gaussian distributions. Each model thus only consists in $NK$ parameters to optimise. This is a simplistic way of describing the most flexible deep neural network. We then maximised on the one hand the KL-OvA (MI) and on the other hand the MMD-OvA. Both clustering results can be seen in Figure 2. We concluded that

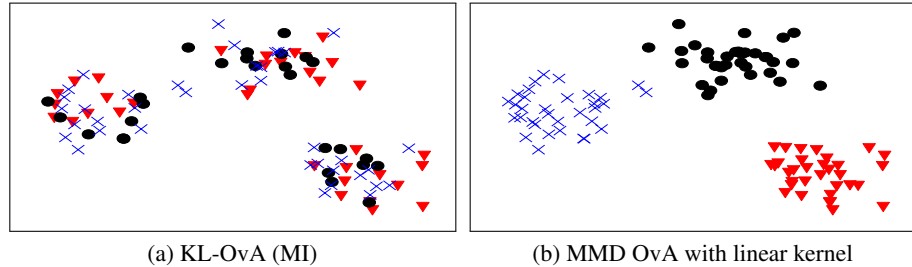

(a) KL-OvA (MI)  (b) MMD OvA with linear kernel

Figure 2: Clustering of a mixture of 3 Gaussian distributions with MI (left) and a GEMINI (right) using categorical distributions. The MI does not have insights on the data shape because of the model, and clusters points uniformly between the 3 clusters (black dots, red triangles and blue crosses) whereas the MMD is aware of the data shape through its kernel.

without any function, e.g. a neural network, to link the parameters of the conditional distribution with $\boldsymbol{x}$, the MI struggles to find the correct decision boundaries. Indeed, the position of $\boldsymbol{x}$ in the 2D space plays no role and the decision boundary becomes only relevant with regards to cluster entropy maximisation: a uniform distribution between 3 clusters. However, it plays a major role in the kernel of the MMD-GEMINI thus solving correctly the problem.

## 4.2 Resistance to outliers

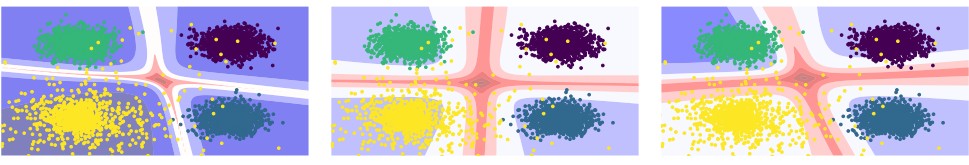

(a) KL-OvA (MI) Entropy Map  (b) MMD-OvO Entropy Map  (c) Wasserstein-OvO Entropy Map

Figure 3: Entropy maps of the predictions of each MLP trained using a GEMINI or the MI. The bottom-left distribution (yellow) is a Student-t distribution with 1 degree of freedom that produces samples far from the origin. The Rényi entropy of prediction is highlighted from lowest (red background) to highest (blue background). MI on the left has the most confident predictions overall and the smallest uncertainty around the decision boundary, i.e. high entropy variations.

To prove the strength of using neural networks for clustering trained with GEMINI, we introduced extreme samples in Gaussian mixtures by replacing a Gaussian distribution with a Student-t distribution for which the degree of freedom $\rho$ is small. We fixed $K = 4$ clusters, 3 being drawn from multivariate Gaussian distributions and the last one from a multivariate Student-t distribution in 2 dimensions for visualisation purposes with 1 degree of freedom (see AppendixE for other parameters and results). Thus, the Student-t distribution has an undefined expectation and produces samples that can be perceived as outliers regarding a Gaussian mixture owing to its heavy tail. We report the ARIs of multi-layered perceptron (MLP) trained 20 times with GEMINIs in Table 2. The presence of "outliers" leads K-Means and Gaussian Mixture models to fail at grasping the 4 distributions when tring to find 4 clusters. Meanwhile, GEMINIs perform better. Note that all MMD and Wasserstein-OvO-GEMINI present lower standard deviation for high scores compared to $f$-divergence GEMINIs. We attribute these performances to both the MLP that tries to find separating hyperplanes in the data space and the absence of hypotheses regarding the data. Moreover, as mentioned in section 2.3, the usual MI is best maximised when its decision boundary presents little entropy $\mathcal{H}(p_\theta(y|\boldsymbol{x}))$. As neural networks can be overconfident (Guo et al., 2017), MI is likely to yield overconfident clustering by minimizing the conditional entropy. We highlight such behaviour in Figure 3 where the Rényi entropy (Rényi, 1961) associated to each sample in the MI (Figure 3a) is much lower, if not 0, compared to MMD-OvO and Wasserstein-OvO (figures 3b and 3c). We conclude that Wasserstein- and MMD-GEMINIs train neural networks not to be overconfident, hence yielding more moderate distributions $p_\theta(y|\boldsymbol{x})$.

Table 2: Mean ARI (std) of a MLP fitting a mixture of 3 Gaussian and 1 Student-t multivariate distributions compared with Gaussian Mixture Models and K-Means. The models try to find 4 at best and the Student-t distribution has $\rho$=1 degree of freedom. We write the ARI for the maximum a posteriori of an oracle aware of all parameters of the data.

| K-Means | GMM | | MMD | | Wasserstein | |
|---|---|---|---|---|---|---|
| | full cov | diagonal cov | $\mathcal{I}_{\mathrm{MMD}}^{\mathrm{ova}}$ | $\mathcal{I}_{\mathrm{MMD}}^{\mathrm{ovo}}$ | $\mathcal{I}_{\mathcal{W}}^{\mathrm{ova}}$ | $\mathcal{I}_{\mathcal{W}}^{\mathrm{ovo}}$ |
| 0 | 0 | 0.024 | 0.922 | 0.921 | 0.915 | 0.922 |
| (0) | (0) | (0.107) | (0.004) | (0.007) | (0.131) | (0.006) |

| Oracle | $f$-divergences | | | | | |
|---|---|---|---|---|---|---|
| | $\mathcal{I}_{\mathrm{KL}}^{\mathrm{ova}}$ | $\mathcal{I}_{\mathrm{KL}}^{\mathrm{ovo}}$ | $\mathcal{I}_{\mathrm{H^2}}^{\mathrm{ova}}$ | $\mathcal{I}_{\mathrm{H^2}}^{\mathrm{ovo}}$ | $\mathcal{I}_{\mathrm{TV}}^{\mathrm{ova}}$ | $\mathcal{I}_{\mathrm{TV}}^{\mathrm{ovo}}$ |
| 0.989 | **0.939** | 0.723 | 0.906 | 0.858 | 0.904 | **0.938** |
| | **(0.006)** | (0.114) | (0.103) | (0.143) | (0.104) | **(0.005)** |

### 4.3 Leveraging a manifold geometry

We highlighted that MI can be maximised without requiring to find the suitable decision boundary. Here, we show how the provided distance to the Wasserstein-OvO GEMINI can leverage appropriate clustering when we have a good a priori on the data.

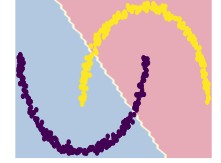 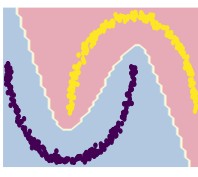

(a) $\mathcal{I}_{kl}^{ova}$ for 2 clusters (MI)  (b) $\mathcal{I}_{\mathcal{W}}^{ovo}$ for 2 clusters

Figure 4: Fitting hand-generated moons using the GEMINI on top of an MLP with 2 clusters to find.

**The importance of the distance $c$:** We generated a dataset consisting of two facing moons on which we trained a MLP using either the MI or the Wasserstein-OvO GEMINI. To construct a distance $c$ for the Wasserstein distance, we derived a distance from the Floyd-Warshall algorithm (Warshall, 1962; Roy, 1959) on a sparse graph describing neighborhoods of samples. This distance describes how many neighbors are in between two samples, further details are provided in appendix H. We report the different decision boundaries in Figure 4. We observe that the insight on the neighborhood provided by our distance $c$ helped the MLP to converge to the correct solution with an appropriate decision boundary unlike the MI. Note that the usual Euclidean distance in the Wasserstein metric would have converged to a solution similar to the MI. Indeed for 2 clusters, the optimal transport plan has a larger value, using a distribution similar to Figure 4a, than in Figure 4b. This toy example shows how an insightful metric provided to the Wasserstein distance in GEMINIs can lead to correct decision boundaries while only designing a discriminative distribution $p_\theta(y|\boldsymbol{x})$ and a distance $c$.

**Not using all clusters** In addition, we highlight an interesting behaviour of all GEMINIs. During optimisation, it is possible that the model converges to using fewer clusters than the number to find. For example in Figure 5, for 5 clusters, the model can converge to 4 balanced clusters and 1 empty cluster (Figure 5b) unlike MI that produced 5 misplaced clusters (Figure 5a). Indeed, the entropy on the cluster proportion in the MI forces to use the maximum number of clusters.

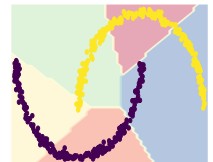 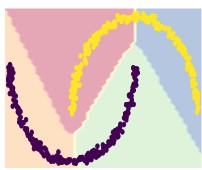

(a) $\mathcal{I}_{kl}^{ova}$ for 5 clusters (MI)  (b) $\mathcal{I}_{\mathcal{W}}^{ovo}$ for 5 clusters

Figure 5: The Wasserstein-ovo model with 5 clusters eventually found 4 unlike the MI that maintained 5 clusters.

Table 3: ARI for deep neural network trained with GEMINIs on MNIST for 500 epochs. Models were trained either with either 10 clusters to find or 15. We indicate in parentheses the number of used clusters by the model after training.

| GEMINI | | 10 clusters | | 15 clusters | |
|---|---|---|---|---|---|
| | | MLP | LeNet-5 | MLP | LeNet-5 |
| KL | OvA | 0.320 (10) | 0.138 (8) | 0.271 (15) | 0.136 (12) |
| | OvO | 0.348 (7) | 0.123 (4) | 0.333 (8) | 0.104 (4) |
| Squared Hellinger | OvA | 0.301 (10) | 0.207 (6) | 0.224 (13) | 0.162 (7) |
| | OvO | 0.287 (10) | 0.161 (6) | 0.305 (13) | 0.157 (7) |
| TV | OvA | 0.299 (10) | 0.171 (6) | 0.277 (15) | 0.140 (6) |
| | OvO | 0.422 (10) | 0.161 (9) | 0.330 (15) | 0.182 (14) |
| MMD | OvA | 0.373 (10) | 0.382 (10) | 0.345 (15) | 0.381 (15) |
| | OvO | 0.361 (10) | 0.373 (10) | 0.364 (15) | 0.379 (15) |
| Wasserstein | OvA | **0.471** (10) | **0.463** (10) | 0.390 (15) | **0.446** (11) |
| | OvO | 0.450 (10) | 0.383 (10) | **0.415** (15) | 0.414 (15) |
| K-Means | | 0.367 | | 0.385 | |

## 4.4 Fitting MNIST

We trained a neural network using either MI or GEMINIs. Following Hu et al. (2017), we first tried with a MLP with one single hidden layer of dimension 1200. To further illustrate the robustness of the method and its adaptability to other architectures, we also experimented using a LeNet-5 architecture (LeCun et al., 1998) since it is adequate to the MNIST dataset. We report our results in Table 3. Since we are dealing with a clustering method, we may not know the number of clusters a priori in a dataset. The only thing that can be said about MNIST is that there are *at least* 10 clusters, one per digit. Indeed, writings of digits could differ leading to more clusters than the number of classes. That is why we further tested the same method with 15 clusters to find in Table 3. We first see that the scores of the MMD and Wasserstein GEMINIs are greater than the MI, with the highest performances for the Wasserstein-OvAWe also observe that no $f$-divergence-GEMINI always yield best ARIs. Nonetheless, we observe better performances in the case of the TV GEMINIs owing to its bounded gradient. This results in controlled stepsize when doing gradient descent contrarily to KL- and squared Hellinger-GEMINIs. Notice that the change of architecture from a MLP to a LeNet-5 unexpectedly halves the scores for the $f$-divergences. We believe this drop is due to the change of notion of neighborhood implied by the network architecture.

## 4.5 Cifar10 clustering using a SIMCLR-derived kernel

To further illustrate the benefits of the kernel or distance provided to GEMINIs, we continue the same experiment as in section 4.4. However, we focus this time on the CIFAR10 dataset. As improved distance, we chose a linear kernel and $\ell_2$ norm between features extracted from a pretrained SIMCLR model (Chen et al., 2020). We provide results for two different architectures: LeNet-5 and ResNet-18 both trained from scratch on raw images, the latter being a common choice of models in deep clustering literature (Van Gansbeke et al., 2020; Tao et al., 2021). We report the results in Table 4 and provide the baseline of MI. We also write the baselines from related works when not using data augmentations to make a fair comparison. Indeed, models trained with GEMINIs do not use data augmentation: only the architecture and the kernel or distance function in the data space plays a role. We observe here that the choice of kernel or distance can be critical in GEMINIs. Indeed, while the Euclidean norm between images does not provide insights on how images of cats and dogs are far as shown by K-Means, features derived from SIMCLR carry much more insight on the proximity of images. This shows that the performances of GEMINIs depend on the quality of distance functions. Interestingly, we observe that for the Resnet-18 using SIMCLR features to guide GEMINIs was not as successful as it has been on the LeNet-5. We believe that the ability of this network to draw any decision boundary makes it equivalent to a categorical distribution model as in Sec. 4.1. Finally, to the best of our knowledge, we are the first to train from scratch a standard discriminative neural network on CIFAR raw images without using labels or direct data augmentations, while getting sensible clustering results. However, other recent methods achieve best scores using data augmentations which we do not (Park et al., 2021).

Table 4: ARI score of models trained for 200 epochs on CIFAR10 with different architectures using GEMINIs. The kernel for the MMD is either a linear kernel or the dot product between features extracted from a pretrained SIMCLR model. Both the Euclidean norm between images and SIMCLR features are considered for the Wasserstein metric. We report the ARI of related works when not using data augmentation for comparison.*: scores reported from Li et al., (2021)

| Architecture | No kernel | Linear kernel / $\ell_2$ norm | | | | SIMCLR (Chen et al., 2020) | | | |
|---|---|---|---|---|---|---|---|---|---|
| | $\mathcal{I}_{\mathrm{KL}}^{\mathrm{ova}}$ | $\mathcal{I}_{\mathrm{MMD}}^{\mathrm{ova}}$ | $\mathcal{I}_{\mathrm{MMD}}^{\mathrm{ovo}}$ | $\mathcal{I}_{\mathcal{W}}^{\mathrm{ova}}$ | $\mathcal{I}_{\mathcal{W}}^{\mathrm{ovo}}$ | $\mathcal{I}_{\mathrm{MMD}}^{\mathrm{ova}}$ | $\mathcal{I}_{\mathrm{MMD}}^{\mathrm{ovo}}$ | $\mathcal{I}_{\mathcal{W}}^{\mathrm{ova}}$ | $\mathcal{I}_{\mathcal{W}}^{\mathrm{ovo}}$ |
| LeNet-5 | 0.026 | 0.049 | 0.048 | 0.043 | 0.041 | **0.157** | 0.145 | 0.079 | 0.138 |
| Resnet-18 | 0.008 | 0.047 | 0.044 | 0.037 | 0.036 | 0.122 | **0.145** | 0.052 | 0.080 |
| KMeans (images / SIMCLR) | | 0.041 | 0.147 | | | CC (Li et al., 2021) | | 0.030 | |
| IDFD (Tao et al., 2021) | | | 0.060 | | | JULE (Yang et al., 2016)* | | 0.138 | |

## 5 Conclusion

We highlighted that the choice of distance at the core of MI can alter the performances of deep learning models when used as an objective for a deep discriminative clustering. We first showed that MI maximisation does not necessarily reflect the best decision boundary in clustering. We introduced the GEMINI, a method which only needs the specification of a neural network and a kernel or distance in the data space. Moreover, we showed how the notion of neighborhood built by the neural network can affect the clustering, especially for MI. To the best of our knowledge, this is the first method that trains single-stage neural networks from scratch using neither data augmentations nor regularisations, yet achieving good clustering performances. We emphasised that GEMINIs are only searching for a maximum number of clusters: after convergence some may be empty. However, we do not have insights to explain this convergence which is part of future work. Finally, we introduced several versions of GEMINIs and would encourage the MMD-OvA or Wasserstein-OvA as a default choice, since it proves to both incorporate knowledge from the data using a kernel or distance while remaining the less complex than MMD-OvO and Wasserstein-OvO in time and memory. OvO versions could be privileged for fine-tuning steps. Future works could include an optimisation of the time performances of the Wasserstein-OvO to make it more competitive.

## Acknowledgements

This work has been supported by the French government, through the 3IA Côte d'Azur, Investment in the Future, project managed by the National Research Agency (ANR) with the reference number ANR-19-P3IA-0002. We would also like to thank the France Canada Research Fund (FFCR) for their contribution to the project. This work was partly supported by EU Horizon 2020 project AI4Media, under contract no. 951911. The authors are grateful to the OPAL infrastructure from Université Côte d'Azur for providing resources and support.

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
