# A  Demonstration of the convergence to 0 of the MI for a Gaussian Mixture

## A.1  Models definition

Let us consider a mixture of two Gaussian distributions, both with different means $\mu_0$ and $\mu_1$, s.t. $\mu_0 < \mu_1$ and of same standard deviation $\sigma$:

$$p(x|y = 0) = \mathcal{N}(x|\mu_0, \sigma^2), p(x|y = 1) = \mathcal{N}(x|\mu_1, \sigma^2),$$

where $y$ is the cluster assignment. We take balanced clusters proportions, i.e. $p(y = 0) = p(y = 1) = \frac{1}{2}$. This first model is the basis that generated the complete dataset $p(x)$. When performing clustering with our discriminative model, we are not aware of the distribution. Consequently: we create other models. We want to compute the difference of mutual information between two decision boundaries that discriminative models $p_\theta(y|x)$ may yield.

We define two decision boundaries: one which splits evenly the data space called $p_A$ and another which splits it on a closed set $p_B$:

$$p_A(y = 1|x) = \left\{ \begin{array}{cc} 1 - \epsilon & x > \frac{\mu_1 - \mu_0}{2} \\ \epsilon & \text{otherwise} \end{array} \right. ,$$

$$p_B(y = 1|x) = \left\{ \begin{array}{cc} 1 - \epsilon & x \in [\mu_0, \mu_1] \\ \epsilon & \text{otherwise} \end{array} \right. . \tag{12}$$

Our goal is to show that both models $p_A$ and $p_B$ will converge to the same value of mutual information as $\epsilon$ converges to 0.

## A.2  Computing cluster proportions

### A.2.1  Cluster proportion of the correct decision boundary

To compute the cluster proportions, we estimate with samples $x$ from the distribution $p_{\text{data}}(x)$. Since we are aware for this demonstration of the true nature of the data distribution, we can use $p(x)$ for sampling. Consequently, we can compute the two marginals:

$$p_A(y = 1) = \int_{\mathcal{X}} p(x)p_A(y = 1|x)dx,$$

$$= \int_{-\infty}^{\frac{\mu_1 - \mu_0}{2}} p(x)\epsilon dx + \int_{\frac{\mu_1 - \mu_0}{2}}^{+\infty} p(x)(1 - \epsilon)dx,$$

$$= \epsilon \left( \int_{-\infty}^{\frac{\mu_1 - \mu_0}{2}} p(x)dx \right) + (1 - \epsilon) \left( \int_{\frac{\mu_1 - \mu_0}{2}}^{+\infty} p(x)dx \right),$$

$$= \frac{1}{2}.$$

### A.2.2  Cluster proportion of the misplaced decision boundary

For the misplaced decision boundary, the marginal is different:

$$p_B(y = 1) = \int_{\mathcal{X}} p(x)p_B(y = 1|x)dx,$$

$$= \epsilon \left( \int_{-\infty}^{\mu_0} p(x)dx + \int_{\mu_1}^{+\infty} p(x)dx \right) + (1 - \epsilon) \int_{\mu_0}^{\mu_1} p(x)dx, \tag{13}$$

$$= \epsilon \left( 1 - \int_{\mu_0}^{\mu_1} p(x)dx \right) + (1 - \epsilon) \int_{\mu_0}^{\mu_1} p(x)dx.$$

Here, we simply introduce a new variable named $\beta$ that will be a shortcut for noting the proportion of data between $\mu_0$ and $\mu_1$:

$$\beta = \int_{\mu_0}^{\mu_1} p(x)dx.$$

And so can we simply write the cluster proportion of decision boundary model B as:

$$p_B(y=1) = \epsilon(1-\beta) + (1-\epsilon)\beta,$$

Leading to the summary of proportions in Table 5. For convenience, we will write the proportions of model B using the shortcuts:

$$\pi_B = p_B(y=1) = \epsilon + \beta(1-2\epsilon),$$

$$\bar{\pi}_B = p_B(y=0) = 1 - \epsilon - \beta(1-2\epsilon).$$

Table 5: Proportions of clusters for models A and B

| $\mathcal{M}$ | A | B |
|---|---|---|
| $p_{\mathcal{M}}(y=1)$ | $\frac{1}{2}$ | $\epsilon + \beta(1-2\epsilon)$ |
| $p_{\mathcal{M}}(y=0)$ | $\frac{1}{2}$ | $1 - \epsilon - \beta(1-2\epsilon)$ |

### A.3 Computing the KL divergences

### A.3.1 Correct decision boundary

We first start by computing the Kullback-Leibler divergence for some arbitrary value of $x \in \mathbb{R}$:

$$D_{\text{KL}}(p_A(y|x)||p_A(y)) = \sum_{i=0}^{1} p_A(y=i|x) \log \frac{p_A(y=i|x)}{p_A(y=i)}.$$

We now need to detail the specific cases, for the value of $p(y=i|x)$ is dependent on $x$. We start $\forall x < \frac{\mu_1 - \mu_0}{2}$:

$$D_{\text{KL}}(p_A(y|x)||p_A(y)) = p_A(y=0|x) \log \frac{p_A(y=0|x)}{\frac{1}{2}} + p_A(y=1|x) \log \frac{p_A(y=1|x)}{\frac{1}{2}},$$

$$= (1-\epsilon) \log 2(1-\epsilon) + \epsilon \log 2\epsilon.$$

The opposite case, $\forall x \geq \frac{\mu_1 - \mu_0}{2}$ yields:

$$D_{\text{KL}}(p_A(y|x)||p_A(y)) = p_A(y=0|x) \log \frac{p_A(y=0|x)}{\frac{1}{2}} + p_A(y=1|x) \log \frac{p_A(y=1|x)}{\frac{1}{2}},$$

$$= \epsilon \log 2\epsilon + (1-\epsilon) \log 2(1-\epsilon).$$

Since both cases are equal, we can write down:

$$D_{\text{KL}}(p_A(y|x)||p_A(y)) = \epsilon \log 2\epsilon + (1-\epsilon) \log 2(1-\epsilon), \forall x \in \mathbb{R}. \tag{14}$$

### A.3.2 Misplaced boundary

We proceed to the same detailing of the Kullback-Leibler divergence computation for the misplaced decision boundary. We start with the set $x \in [\mu_0, \mu_1]$:

$$D_{\mathrm{KL}}(p_B(y|x)||p_B(y)) = p_B(y=0|x) \log \frac{p_B(y=0|x)}{p_B(y=0)} + p_B(y=1|x) \log \frac{p_B(y=1|x)}{p_B(y=1)},$$

$$= \epsilon \log \frac{\epsilon}{\bar{\pi}_B} + (1-\epsilon) \log \frac{1-\epsilon}{\pi_B}.$$

When $x$ is out of this set, the divergence becomes:

$$D_{\mathrm{KL}}(p_B(y|x)||p_B(y)) = p_B(y=0|x) \log \frac{p_B(y=0|x)}{p_B(y=0)} + p_B(y=1|x) \log \frac{p_B(y=1|x)}{p_B(y=1)},$$

$$= (1-\epsilon) \log \frac{1-\epsilon}{\bar{\pi}_B} + \epsilon \log \frac{\epsilon}{\pi_B}.$$

To fuse the two results, we will write the KL divergence as such:

$$D_{\mathrm{KL}}(p_B(y|x)||p_B(y)) = \epsilon \log \epsilon + (1-\epsilon) \log(1-\epsilon) - C(x), \forall x \in \mathbb{R}, \tag{15}$$

where $C(x)$ is a constant term depending on $x$ defined by:

$$C(x) = \begin{cases} \epsilon \log \bar{\pi}_B + (1-\epsilon) \log \pi_B & x \in [\mu_0, \mu_1] \\ \epsilon \log \pi_B + (1-\epsilon) \log \bar{\pi}_B & x \in \mathbb{R} \setminus [\mu_0, \mu_1] \end{cases}. \tag{16}$$

For simplicity of later writings, we will shorten the notations by:

$$C(x) = \begin{cases} \alpha_1 & x \in [\mu_0, \mu_1] \\ \alpha_0 & x \in \mathbb{R} \setminus [\mu_0, \mu_1] \end{cases}.$$

## A.4 Evaluating the mutual information

### A.4.1 Correct decision boundary

We inject the value of the Kullback-Leibler divergence from Eq. (14) inside an expectation performed over the data distribution $p_{\mathrm{data}}(x)$:

$$\mathcal{I}_A(X;Y) = \mathbb{E}_{x \sim p_{\mathrm{data}}(x)} \left[ D_{\mathrm{KL}}(p_A(y|x)||p_A(y)) \right], \tag{17}$$

$$= \int_{\mathcal{X}} p(x) \left( \epsilon \log(2\epsilon) + (1-\epsilon) \log(2(1-\epsilon)) \right) dx, \tag{18}$$

$$= \epsilon \log(2\epsilon) + (1-\epsilon) \log(2(1-\epsilon)). \tag{19}$$

Since the KL divergence was independent of $x$, we could leave the constant outside of the integral which is equal to 1.

We can assess the coherence of Eq. (19) since its limit as $\epsilon$ approaches 0 is $\log 2$. In terms of bits, this is the same as saying that the information on $X$ directly gives us information on the $Y$ of the cluster.

### A.4.2 Odd decision boundary

We inject the value of the KL divergence from Eq. (15) inside the expectation of the mutual information:

$$\mathcal{I}_B(X;Y) = \mathbb{E}_{x \sim p_{\text{data}}(x)} \left[ D_{\text{KL}}(p_B(y|x) || p_B(y)) \right],$$

$$= \int_{\mathcal{X}} p(x) \left( \epsilon \log \epsilon + (1-\epsilon) \log(1-\epsilon) - C(x) \right), dx$$

$$= \epsilon \log \epsilon + (1-\epsilon) \log(1-\epsilon) - \int_{\mathcal{X}} p(x) C(x) dx.$$

The first terms are constant with respect to $x$ and the integral of $p(x)$ over $\mathcal{X}$ adds up to 1. We finally need to detail the expectation of the constant $C(x)$ from Eq. (16):

$$\mathbb{E}_x[C(x)] = \int_{-\infty}^{\mu_0} C(x) p(x) dx + \int_{\mu_0}^{\mu_1} C(x) p(x) dx + \int_{\mu_1}^{+\infty} C(x) p(x) dx,$$

$$= \alpha_0 \left( \int_{-\infty}^{\mu_0} p(x) dx + \int_{\mu_1}^{+\infty} p(x) dx \right) + \alpha_1 \int_{\mu_0}^{\mu_1} p(x) dx,$$

$$= \alpha_0 (1 - \beta) + \alpha_1 \beta.$$

This can be further improved by unfolding the description of $\alpha_0$ and $\alpha_1$ from Eq. (16):

$$\alpha_0 (1 - \beta) + \beta \alpha_1 = \alpha_0 + \beta(\alpha_1 - \alpha_0),$$

$$= \epsilon \log \pi_B + (1-\epsilon) \log \bar{\pi}_B + \beta \left[ \epsilon \log \bar{\pi}_B + (1-\epsilon) \log \pi_B \right.$$

$$\left. - \epsilon \log \pi_B - (1-\epsilon) \log \bar{\pi}_B \right],$$

$$= [1 - \epsilon + \beta \epsilon - \beta + \beta \epsilon] \log \bar{\pi}_B + [\epsilon + \beta - \beta \epsilon - \beta \epsilon] \log \pi_B,$$

$$= \log \bar{\pi}_B + [2\beta \epsilon - \beta - \epsilon] \log \frac{\bar{\pi}_B}{\pi_B}.$$

We can finally write down the mutual information for the model B:

$$\mathcal{I}_B(X;Y) = \epsilon \log \epsilon + (1-\epsilon) \log(1-\epsilon) - \log \bar{\pi}_B - [2\beta \epsilon - \beta - \epsilon] \log \frac{\bar{\pi}_B}{\pi_B}. \tag{20}$$

### A.5 Differences of mutual information

Now that we have the exact value of both mutual informations, we can compute their differences:

$$\Delta_{\mathcal{I}} = \mathcal{I}_A(X;Y) - \mathcal{I}_B(X;Y),$$

$$= \epsilon \log(2\epsilon) + (1-\epsilon) \log(2(1-\epsilon)) - \epsilon \log \epsilon - (1-\epsilon) \log(1-\epsilon)$$

$$+ \log \bar{\pi}_B + [2\beta \epsilon - \beta - \epsilon] \log \frac{\bar{\pi}_B}{\pi_B},$$

$$= \epsilon \log 2 + (1-\epsilon) \log 2 + \log \bar{\pi}_B + [2\beta \epsilon - \beta - \epsilon] \log \frac{\bar{\pi}_B}{\pi_B}.$$

We then deduce how the difference of mutual information evolves as the decision boundary becomes sharper, i.e. when $\epsilon$ approaches 0:

$$\lim_{\epsilon \to 0} \Delta_{\mathcal{I}} = \log 2 + \log \bar{\pi}_B - \beta \log \frac{\bar{\pi}_B}{\pi_B}.$$

However, the cluster proportions by B $\pi_B$ also take a different value as $\epsilon$ approaches 0. Recalling Eq. (12):

$$\lim_{\epsilon \to 0} p_B(y = 1) = \beta, \lim_{\epsilon \to 0} p_B(y = 0) = 1 - \beta.$$

And finally can we write that:

$$\lim_{\epsilon \to 0} \Delta_{\mathcal{I}} = \log 2 + \log(1 - \beta) - \beta \log \frac{1 - \beta}{\beta},$$
$$= \log 2 + (1 - \beta) \log(1 - \beta) + \beta \log \beta,$$
$$= \log 2 - \mathcal{H}(\beta). \tag{21}$$

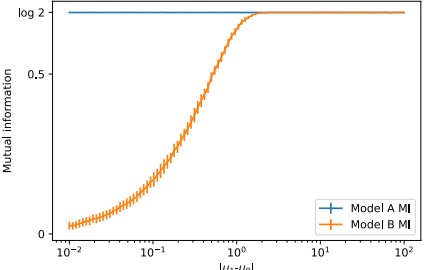
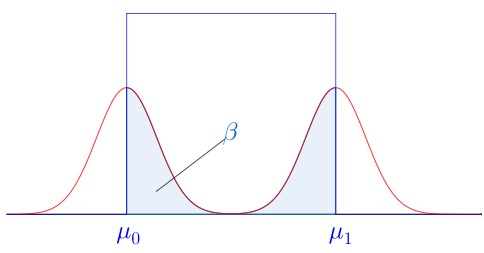

(a) Differences of MI between models A and B   (b) Gaussian mixture distribution $p(x)$ with proportion $\beta$ in between the two means $\mu_0$ and $\mu_1$

Figure 6: Value of the mutual information for the two models splitting a Gaussian mixture depending on the distance between the two means $\mu_0$ and $\mu_1$ of the two generating Gaussian distributions. We estimate the MI by computing it 50 times on 1000 samples drawn from the Gaussian mixture.

To conclude, as the decision boundaries turn sharper, i.e. when $\epsilon$ approaches 0, the difference of mutual information between the two models is controlled by the entropy of proportion of data $\beta$ between the two means $\mu_0$ and $\mu_1$. We know that for binary entropies, the optimum is reached for $\beta = 0.5$. In other words having $\mu_0$ and $\mu_1$ distant enough to ensure balance of proportions between the two clusters of model B leads to a difference of mutual information equal to 0. We experimentally highlight this convergence in Figure 6 where we compute the mutual information of models A and B as the distance between the two means $\mu_0$ and $\mu_1$ increases in the Gaussian distribution mixture.

## B   A geometrical perspective on OvA and OvO GEMINIs

Considering the topology of the data space through a kernel in the case of the MMD or a distance in the case of the Wasserstein metric implies that we can effectively measure how two distributions are close to another. In the formal design of the mutual information, the distribution of each cluster

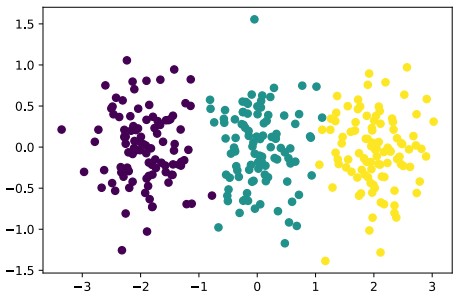

Figure 7: Here, 3 clusters of equal proportions from isotropic Gaussian distributions are located in -2, 0 and 2 on the x-axis, with small covariance. The complete data distribution hence has its expectation in 0 on the x-axis. Consequently, maximising the MMD OvA GEMINI can lead to 2 clusters at best while the MMD OvO is able to see all 3 clusters.

$p(\boldsymbol{x}|y)$ is compared to the complete data distribution $p(\boldsymbol{x})$. Therefore, if one distribution of a specific cluster $p(\boldsymbol{x}|y)$ were to look alike the data distribution $p(\boldsymbol{x})$, for example up to a constant in some areas of the space, then its distance to the data distribution could be 0, making it unnoticed when maximising the OvA GEMINI.

Take the example of 3 distributions $\{p(\boldsymbol{x}|y=i)\}_{i=1}^{3}$ with respective different expectations $\{\mu_i\}_{i=1}^{3}$. We want to separate them using the MMD OvA GEMINI with linear kernel. The mixture of the 3 distributions creates a data distribution with expectation $\mu = \sum_{i=1}^{3} p(y=i)\mu_i$. However if the distributions satisfy that this data expectation $\mu$ is equal to one of the sub-expectations $\mu_i$, then the associated distribution $i$ will be non-evaluated since its MMD to the data distribution is equal to 0. We illustrate this example in figure 7. We tackled the problem by introducing the OvO setup where each pair of different cluster distribution is compared.

## C    Another approach to Wasserstein maximisation

The Wasserstein-1 metric can be considered as an IPM defined over a set of 1-Lipschitz functions. Indeed, such writing is the dual representation of the Wasserstein-1 metric:

$$W_c(p\|q) = \sup_{f, \|f\|_L \leq 1} \mathbb{E}_{x \sim p}[f(x)] - \mathbb{E}_{z \sim q}[f(z)].$$

Yet, evaluating a supremum as an objective to maximise is hardly compatible with the usual back-propagation in neural networks. This definition was used in attempts to stabilise GAN training (Arjovsky et al., 2017) by using 1-Lipschitz neural networks (Gouk et al., 2021). However, the Lipschitz continuity was achieved at the time by weight clipping, whereas other methods such as spectral normalisation (Miyato et al., 2018a) now allow arbitrarily large weights. The restriction of 1-Lipscthiz functions to 1-Lipschitz neural networks does not equal the true Wasserstein distance, and the term "neural net distance" is sometimes preferred (Arora et al., 2017). Still, estimating the Wasserstein distance using a set of Lipschitz functions derived from neural networks architectures brings more difficulties to actually leverage the true distance according to the energy cost $c$ of Eq. 10.

Globally, we hardly experimented the generic IPM for GEMINIs. Our efforts for defining a set of 1-Lipschitz critics, one per cluster for OvA or one per pair of clusters for OvO, to perform the neural net distance (Arora et al., 2017) were not fruitful. This is mainly because such objective requires the definition of one neural network for the posterior distribution $p_\theta(y|\boldsymbol{x})$ and $K$ (resp. $K(K-1)/2$) other 1-Lipschitz neural networks for the OvA (resp. OvO) critics, i.e. a large amount of parameters. Moreover, this brings the problem of designing not only one, but many neural networks while the design of one accurate architecture is already a sufficient problem.

## D    Deriving GEMINIs

We show in this appendix how to derive all estimable forms of the GEMINI.

### D.1    $f$-divergence GEMINI

We detail here the derivation for 3 $f$-divergences that we previously chose: the KL divergence, the TV distance and the squared Hellinger distance, as well as the generic scenario for any function $f$.

#### D.1.1   Generic scenario

First, we recall that the definition of an $f$-divergence involves a convex function:

$$f : \mathbb{R}^+ \to \mathbb{R}$$
$$x \to f(x)$$
$$\text{s.t.} \quad f(1) = 0,$$

between two distributions $p$ and $q$ as described:

$$D_{\text{f-div}}(p, q) = \mathbb{E}_{\boldsymbol{x} \sim q}\left[ f\left( \frac{p(\boldsymbol{x})}{q(\boldsymbol{x})} \right) \right].$$

We simply inject this definition in the OvA-GEMINI and directly obtain both an expectation on the cluster assignment $y$ and on the data variable $\boldsymbol{x}$. We then merge the writing of the two expectations for the sake of clarity.

$$\begin{aligned}
\mathcal{I}_{\text{f-div}}^{\text{ova}} &= \mathbb{E}_{p_\theta(y)}\left[ D_{\text{f-div}}(p_\theta(\boldsymbol{x}|y) || p_{\text{data}}(\boldsymbol{x})) \right], \\
&= \mathbb{E}_{p_\theta(y)}\left[ \mathbb{E}_{p_{\text{data}}(\boldsymbol{x})}\left[ f\left( \frac{p_\theta(\boldsymbol{x}|y)}{p_{\text{data}}(\boldsymbol{x})} \right) \right] \right], \\
&= \mathbb{E}_{p_\theta(y), p_{\text{data}}(\boldsymbol{x})}\left[ f\left( \frac{p_\theta(y|\boldsymbol{x})}{p_\theta(y)} \right) \right].
\end{aligned}$$

Injecting the $f$-divergence in the OvO-GEMINI first yields:

$$\begin{aligned}
\mathcal{I}_{\text{f-div}}^{\text{ovo}} &= \mathbb{E}_{p_\theta(y_a), p_\theta(y_b)}\left[ D_{\text{f-div}}(p_\theta(\boldsymbol{x}|y_a) || p_\theta(\boldsymbol{x}|y_b)) \right], \\
&= \mathbb{E}_{p_\theta(y_a), p_\theta(y_b)}\left[ \mathbb{E}_{p_\theta(\boldsymbol{x}|y_b)}\left[ f\left( \frac{p_\theta(\boldsymbol{x}|y_a)}{p_\theta(\boldsymbol{x}|y_b)} \right) \right] \right].
\end{aligned}$$

Now, by using Bayes theorem, we can perform the inner expectation over the data distribution. We then merge the expectations for the sake of clarity.

$$\begin{aligned}
\mathcal{I}_{\text{f-div}}^{\text{ovo}} &= \mathbb{E}_{p_\theta(y_a), p_\theta(y_b)}\left[ \mathbb{E}_{p_{\text{data}}(\boldsymbol{x})}\left[ \frac{p_\theta(y_b|\boldsymbol{x})}{p_\theta(y_b)} f\left( \frac{p_\theta(\boldsymbol{x}|y_a)}{p_\theta(\boldsymbol{x}|y_b)} \right) \right] \right], \\
&= \mathbb{E}_{p_\theta(y_a), p_\theta(y_b), p_{\text{data}}(\boldsymbol{x})}\left[ \frac{p_\theta(y_b|\boldsymbol{x})}{p_\theta(y_b)} f\left( \frac{p_\theta(y_a|\boldsymbol{x})p_\theta(y_b)}{p_\theta(y_b|\boldsymbol{x})p_\theta(y_a)} \right) \right].
\end{aligned}$$

Notice that we also changed the ratio of conditional distributions inside the function by a ratio of posteriors through Bayes' theorem, weighted by the relative cluster proportions.

Now, we can derive into details these equations for the 3 $f$-divergences we focused on: the KL divergence, the TV distance and the squared Hellinger distance.

### D.1.2 Kullback-Leibler divergence

The function for Kullback-Leibler is $f(t) = t \log t$. We do not need to write the OvA equation: it is straightforwardly the usual MI. For the OvO, we inject the function definition by replacing:

$$t = \frac{p_\theta(y_a|\boldsymbol{x})p_\theta(y_b)}{p_\theta(y_b|\boldsymbol{x})p_\theta(y_a)},$$

in order to get:

$$\mathcal{I}_{\text{KL}}^{\text{ovo}} = \mathbb{E}_{p_\theta(y_a), p_\theta(y_b), p_{\text{data}}(\boldsymbol{x})}\left[ \frac{p_\theta(y_b|\boldsymbol{x})}{p_\theta(y_b)} \times \frac{p_\theta(y_a|\boldsymbol{x})p_\theta(y_b)}{p_\theta(y_b|\boldsymbol{x})p_\theta(y_a)} \log \frac{p_\theta(y_a|\boldsymbol{x})p_\theta(y_b)}{p_\theta(y_b|\boldsymbol{x})p_\theta(y_a)} \right].$$

We can first simplify the factors outside of the logs:

$$\mathcal{I}_{\text{KL}}^{\text{ovo}} = \mathbb{E}_{p_\theta(y_a), p_\theta(y_b), p_{\text{data}}(\boldsymbol{x})} \left[ \frac{p_\theta(y_a|\boldsymbol{x})}{p_\theta(y_a)} \log \frac{p_\theta(y_a|\boldsymbol{x})p_\theta(y_b)}{p_\theta(y_b|\boldsymbol{x})p_\theta(y_a)} \right].$$

If we use the properties of the log, we can separate the inner term in two sub-expressions:

$$\mathcal{I}_{\text{KL}}^{\text{ovo}} = \mathbb{E}_{p_\theta(y_a), p_\theta(y_b), p_{\text{data}}(\boldsymbol{x})} \left[ \frac{p_\theta(y_a|\boldsymbol{x})}{p_\theta(y_a)} \log \frac{p_\theta(y_a|\boldsymbol{x})}{p_\theta(y_a)} + \frac{p_\theta(y_a|\boldsymbol{x})}{p_\theta(y_a)} \log \frac{p_\theta(y_b)}{p_\theta(y_b|\boldsymbol{x})} \right].$$

Hence, we can use the linearity of the expectation to separate the two terms above. The first term is constant w.r.t. $y_b$, so we can remove this variable from the expectation among the subscripts:

$$\mathcal{I}_{\text{KL}}^{\text{ovo}} = \mathbb{E}_{p_\theta(y_a), p_{\text{data}}(\boldsymbol{x})} \left[ \frac{p_\theta(y_a|\boldsymbol{x})}{p_\theta(y_a)} \log \frac{p_\theta(y_a|\boldsymbol{x})}{p_\theta(y_a)} \right] + \mathbb{E}_{p_\theta(y_a), p_\theta(y_b), p_{\text{data}}(\boldsymbol{x})} \left[ \frac{p_\theta(y_a|\boldsymbol{x})}{p_\theta(y_a)} \log \frac{p_\theta(y_b)}{p_\theta(y_b|\boldsymbol{x})} \right].$$

Since the variables $y_a$ and $y_b$ are independent, we can use the fact that:

$$\mathbb{E}_{p_\theta(y_a)} \left[ \frac{p_\theta(y_a|\boldsymbol{x})}{p_\theta(y_a)} \right] = \int p_\theta(y_a) \frac{p_\theta(y_a|\boldsymbol{x})}{p_\theta(y_a)} dy_a = 1,$$

inside the second term to reveal the final form of the equation:

$$\mathcal{I}_{\text{KL}}^{\text{ovo}} = \mathbb{E}_{p_{\text{data}}(\boldsymbol{x}), p_\theta(y)} \left[ \frac{p_\theta(y|\boldsymbol{x})}{p_\theta(y)} \log \frac{p_\theta(y|\boldsymbol{x})}{p_\theta(y)} \right] + \mathbb{E}_{p_{\text{data}}(\boldsymbol{x}), p_\theta(y)} \left[ \log \frac{p_\theta(y)}{p_\theta(y|\boldsymbol{x})} \right].$$

Notice that since both terms did not compare one cluster assignment $y_a$ against another $y_b$, we can switch to the same common variable $y$. Both terms are in fact KL divergences depending on the cluster assignment $y$. The first is the reverse of the second. This sum of KL divergences is sometimes called the *symmetric* KL, and so can we write in two ways the KL-OvO GEMINI:

$$\mathcal{I}_{\text{KL}}^{\text{ovo}} = \mathbb{E}_{p_{\text{data}}(\boldsymbol{x})} \left[ D_{\text{KL}}(p_\theta(y|\boldsymbol{x})||p_\theta(y)) \right] + \mathbb{E}_{p_{\text{data}}(\boldsymbol{x})} \left[ D_{\text{KL}}(p_\theta(y)||p_\theta(y|\boldsymbol{x})) \right],$$
$$= \mathbb{E}_{p_{\text{data}}(\boldsymbol{x})} \left[ D_{\text{KL-sym}}(p_\theta(y|\boldsymbol{x})||p_\theta(y)) \right].$$

We can also think of this equation as the usual MI with an additional term based on the reversed KL divergence.

### D.1.3 Total Variation distance

For the total variation, the function is $f(t) = \frac{1}{2}|t - 1|$. Thus, the OvA GEMINI is:

$$\mathcal{I}_{\text{TV}}^{\text{ova}} = \frac{1}{2} \mathbb{E}_{p_\theta(y), p_{\text{data}}(\boldsymbol{x})} \left[ \left| \frac{p_\theta(y|\boldsymbol{x})}{p_\theta(y)} - 1 \right| \right].$$

And the OvO is:

$$\mathcal{I}_{\text{TV}}^{\text{ovo}} = \frac{1}{2} \mathbb{E}_{p_\theta(y_a), p_\theta(y_b), p_{\text{data}}(\boldsymbol{x})} \left[ \left| \frac{p_\theta(y_b|\boldsymbol{x})}{p_\theta(y_b)} \right| \frac{p_\theta(y_a|\boldsymbol{x})p_\theta(y_b)}{p_\theta(y_b|\boldsymbol{x})p_\theta(y_a)} - 1 \right| \right],$$
$$= \frac{1}{2} \mathbb{E}_{p_\theta(y_a), p_\theta(y_b), p_{\text{data}}(\boldsymbol{x})} \left[ \left| \frac{p_\theta(y_a|\boldsymbol{x})}{p_\theta(y_a)} - \frac{p_\theta(y_b|\boldsymbol{x})}{p_\theta(y_b)} \right| \right].$$

We did not find any further simplification of these equations.

### D.1.4 Squared Hellinger distance

Finally, the squared Hellinger distance is based on $f(t) = 2(1 - \sqrt{t})$. Hence the OvA unfolds as:

$$\mathcal{I}_{\mathrm{H}^2}^{\mathrm{ova}} = \mathbb{E}_{p_\theta(y), p_{\mathrm{data}}(\boldsymbol{x})} \left[ 2 \left( 1 - \sqrt{\frac{p_\theta(y|\boldsymbol{x})}{p_\theta(y)}} \right) \right],$$

$$= 2 - 2\mathbb{E}_{p_{\mathrm{data}}(\boldsymbol{x}), p_\theta(y)} \left[ \sqrt{\frac{p_\theta(y|\boldsymbol{x})}{p_\theta(y)}} \right].$$

The idea of the squared Hellinger-OvA GEMINI is therefore to minimise the expected square root of the relative certainty between the posterior and cluster proportion.

For the OvO setting, the definition yields:

$$\mathcal{I}_{\mathrm{H}^2}^{\mathrm{ovo}} = \mathbb{E}_{p_\theta(y_a), p_\theta(y_b), p_{\mathrm{data}}(\boldsymbol{x})} \left[ \frac{p_\theta(y_b|\boldsymbol{x})}{p_\theta(y_b)} \times 2 \times \left( 1 - \sqrt{\frac{p_\theta(y_a|\boldsymbol{x})p_\theta(y_b)}{p_\theta(y_b|\boldsymbol{x})p_\theta(y_a)}} \right) \right],$$

which we can already simplify by putting the constant 2 outside of the expectation, and by inserting all factors inside the square root before simplifying and separating the expectation:

$$\mathcal{I}_{\mathrm{H}^2}^{\mathrm{ovo}} = 2\mathbb{E}_{p_\theta(y_a), p_\theta(y_b), p_{\mathrm{data}}(\boldsymbol{x})} \left[ \frac{p_\theta(y_b|\boldsymbol{x})}{p_\theta(y_b)} - \frac{p_\theta(y_b|\boldsymbol{x})}{p_\theta(y_b)} \sqrt{\frac{p_\theta(y_a|\boldsymbol{x})p_\theta(y_b)}{p_\theta(y_a)p_\theta(y_b|\boldsymbol{x})}} \right],$$

$$= 2\mathbb{E}_{p_\theta(y_a), p_\theta(y_b), p_{\mathrm{data}}(\boldsymbol{x})} \left[ \frac{p_\theta(y_b|\boldsymbol{x})}{p_\theta(y_b)} \right] - 2\mathbb{E}_{p_\theta(y_a), p_\theta(y_b), p_{\mathrm{data}}(\boldsymbol{x})} \left[ \sqrt{\frac{p_\theta(y_a|\boldsymbol{x})p_\theta(y_b|\boldsymbol{x})}{p_\theta(y_a)p_\theta(y_b)}} \right].$$

We can replace the first term by the constant 1, as shown for the KL-OvO derivation. Since we can split the square root into the product of two square roots, we can apply twice the expectation over $y_a$ and $y_b$ because these variables are independent:

$$\mathcal{I}_{\mathrm{H}^2}^{\mathrm{ovo}} = 2 - 2\mathbb{E}_{p_{\mathrm{data}}(\boldsymbol{x})} \left[ \mathbb{E}_{p_\theta(y)} \left[ \sqrt{\frac{p_\theta(y|\boldsymbol{x})}{p_\theta(y)}} \right]^2 \right].$$

To avoid computing this squared expectation, we use the equation of the variance $\mathbb{V}$ to replace it. Thus:

$$\mathcal{I}_{\mathrm{H}^2}^{\mathrm{ovo}} = 2 - 2\mathbb{E}_{p_{\mathrm{data}}(\boldsymbol{x})} \left[ \mathbb{E}_{p_\theta(y)} \left[ \frac{p_\theta(y|\boldsymbol{x})}{p_\theta(y)} \right] - \mathbb{V}_{p_\theta(y)} \left[ \sqrt{\frac{p_\theta(y|\boldsymbol{x})}{p_\theta(y)}} \right] \right],$$

$$= 2 - 2\mathbb{E}_{p_{\mathrm{data}}(\boldsymbol{x})} \left[ \mathbb{E}_{p_\theta(y)} \left[ \frac{p_\theta(y|\boldsymbol{x})}{p_\theta(y)} \right] \right] + 2\mathbb{E}_{p_{\mathrm{data}}(\boldsymbol{x})} \left[ \mathbb{V}_{p_\theta(y)} \left[ \sqrt{\frac{p_\theta(y|\boldsymbol{x})}{p_\theta(y)}} \right] \right].$$

Then, for the same reason as before, the second term is worth 1, which cancels the first constant. We therefore end up with:

$$\mathcal{I}_{\mathrm{H}^2}^{\mathrm{ovo}} = 2\mathbb{E}_{p_{\mathrm{data}}(\boldsymbol{x})} \left[ \mathbb{V}_{p_\theta(y)} \left[ \sqrt{\frac{p_\theta(y|\boldsymbol{x})}{p_\theta(y)}} \right] \right].$$

Similar to the KL-OvO case, the squared Hellinger OvO converges to an OvA setting, i.e. we only need information about the cluster distribution itself without comparing it to another. Furthermore, the idea of maximising the variance of the cluster assignments is straightforward for clustering.

## D.2 Maximum Mean Discrepancy

When using an IPM with a family of functions that project an input of $\mathcal{X}$ to the unit ball of an RKHS $\mathcal{H}$, the IPM becomes the MMD distance.

$$\begin{aligned} \mathrm{MMD}(p, q) &= \sup_{f:||f||_{\mathcal{H}} \leq 1} \mathbb{E}_{\boldsymbol{x}_a \sim p}[f(\boldsymbol{x}_a)] - \mathbb{E}_{\boldsymbol{x}_b \sim q}[f(\boldsymbol{x}_b)], \\ &= \|\mathbb{E}_{\boldsymbol{x}_a \sim p}[\varphi(\boldsymbol{x}_a)] - \mathbb{E}_{\boldsymbol{x}_b \sim q}[\varphi(\boldsymbol{x}_b)]\|_{\mathcal{H}}, \end{aligned}$$

where $\varphi$ is a embedding function of the RKHS.

By using a kernel function $\kappa(\boldsymbol{x}_a, \boldsymbol{x}_b) = <\varphi(\boldsymbol{x}_a), \varphi(\boldsymbol{x}_b)>$, we can express the square of this distance thanks to inner product space properties (Gretton et al., 2012):

$$\mathrm{MMD}^2(p, q) = \mathbb{E}_{\boldsymbol{x}_a, \boldsymbol{x}_a' \sim p}[\kappa(\boldsymbol{x}_a, \boldsymbol{x}_a')] + \mathbb{E}_{\boldsymbol{x}_b, \boldsymbol{x}_b' \sim q}[\kappa(\boldsymbol{x}_b, \boldsymbol{x}_b')] - 2\mathbb{E}_{\boldsymbol{x}_a \sim p, \boldsymbol{x}_b \sim q}[\kappa(\boldsymbol{x}_a, \boldsymbol{x}_b)].$$

Now, we can derive each term of this equation using our distributions $p \equiv p_\theta(\boldsymbol{x}|y)$ and $q \equiv p_{\mathrm{data}}(\boldsymbol{x})$ for the OvA case, and $p \equiv p_\theta(\boldsymbol{x}|y_a), q \equiv p_\theta(\boldsymbol{x}|y_b)$ for the OvO case. In both scenarios, we aim at finding an expectation over the data variable $x$ using only the respectively known and estimable terms $p_\theta(y|\boldsymbol{x})$ and $p_\theta(y)$.

**OvA scenario** For the first term, we use Bayes' theorem twice to get an expectation over two variables $\boldsymbol{x}_a$ and $\boldsymbol{x}_b$ drawn from the data distribution.

$$\begin{aligned} \mathbb{E}_{\boldsymbol{x}_a, \boldsymbol{x}_a' \sim p} &= \mathbb{E}_{\boldsymbol{x}_a, \boldsymbol{x}_a' \sim p_\theta(\boldsymbol{x}|y)} \left[ \kappa(\boldsymbol{x}_a, \boldsymbol{x}_a') \right], \\ &= \mathbb{E}_{\boldsymbol{x}_a, \boldsymbol{x}_a' \sim p_{\mathrm{data}}(\boldsymbol{x})} \left[ \frac{p_\theta(y|\boldsymbol{x}_a) p_\theta(y|\boldsymbol{x}_a')}{p_\theta(y)^2} \kappa(\boldsymbol{x}_a, \boldsymbol{x}_a') \right]. \end{aligned}$$

For the second term, we do not need to perform anything particular as we direcly get an expectation over the data variabes $\boldsymbol{x}_a$ and $\boldsymbol{x}_b$.

$$\mathbb{E}_{\boldsymbol{x}_b, \boldsymbol{x}_b' \sim q} = \mathbb{E}_{\boldsymbol{x}_b, \boldsymbol{x}_b' \sim p_{\mathrm{data}}(\boldsymbol{x})} \left[ \kappa(\boldsymbol{x}_b, \boldsymbol{x}_b') \right].$$

The last term only needs Bayes theorem once, for the distribution $q$ is directly replaced by the data distribution $p_{\mathrm{data}}(\boldsymbol{x})$:

$$\begin{aligned} \mathbb{E}_{\boldsymbol{x}_a \sim p, \boldsymbol{x}_b \sim q} &= \mathbb{E}_{\boldsymbol{x}_a \sim p_\theta(\boldsymbol{x}|y), \boldsymbol{x}_b \sim p_{\mathrm{data}}(\boldsymbol{x})} \left[ \kappa(\boldsymbol{x}_a, \boldsymbol{x}_b) \right], \\ &= \mathbb{E}_{\boldsymbol{x}_a, \boldsymbol{x}_b \sim px} \left[ \frac{p_\theta(y|\boldsymbol{x}_a)}{p_\theta(y)} \kappa(\boldsymbol{x}_a, \boldsymbol{x}_b) \right]. \end{aligned}$$

Note that for the last term, we could replace $p_\theta(y|\boldsymbol{x}_a)$ by $p_\theta(y|\boldsymbol{x}_b)$; that would not affect the result since $\boldsymbol{x}_a$ and $\boldsymbol{x}_b$ are independently drawn from $p_{\mathrm{data}}(\boldsymbol{x})$. We thus replace all terms, and do not forget to put a square root on the entire sum since we have computed so far the squared MMD:

$$\begin{aligned} \mathcal{I}_{\mathrm{MMD}}^{\mathrm{ova}} &= \mathbb{E}_{p_\theta(y)} \left[ \mathrm{MMD}(p_\theta(\boldsymbol{x}|y), p_{\mathrm{data}}(\boldsymbol{x})) \right], \\ &= \mathbb{E}_{p_\theta(y)} \left[ \left( \mathbb{E}_{\boldsymbol{x}_a, \boldsymbol{x}_a' \sim p_{\mathrm{data}}(\boldsymbol{x})} \left[ \frac{p_\theta(y|\boldsymbol{x}_a) p_\theta(y|\boldsymbol{x}_a')}{p_\theta(y)^2} \kappa(\boldsymbol{x}_a, \boldsymbol{x}_a') \right] \right. \right. \\ &\quad \left. \left. + \mathbb{E}_{\boldsymbol{x}_b, \boldsymbol{x}_b' \sim p_{\mathrm{data}}(\boldsymbol{x})} \left[ \kappa(\boldsymbol{x}_b, \boldsymbol{x}_b') \right] - 2\mathbb{E}_{\boldsymbol{x}_a, \boldsymbol{x}_b \sim p_{\mathrm{data}}(\boldsymbol{x})} \left[ \frac{p_\theta(y|\boldsymbol{x}_a)}{p_\theta(y)} \kappa(\boldsymbol{x}_a, \boldsymbol{x}_b) \right] \right)^{\frac{1}{2}} \right]. \end{aligned}$$

Since all variables $\boldsymbol{x}_a$, $\boldsymbol{x}_a'$, $\boldsymbol{x}_b$ and $\boldsymbol{x}_b'$ are independently drawn from the same distribution $p_{\text{data}}(\boldsymbol{x})$, we can replace all of them by the variables $\boldsymbol{x}$ and $\boldsymbol{x}'$. We then use the linearity of the expectation and factorise by the kernel $\kappa(\boldsymbol{x}, \boldsymbol{x}')$:

$$
\mathcal{I}_{\text{MMD}}^{\text{ova}} = \mathbb{E}_{p_\theta(y)} \left[ \mathbb{E}_{\boldsymbol{x}, \boldsymbol{x}' \sim p_{\text{data}}(\boldsymbol{x})} \left[ \kappa(\boldsymbol{x}, \boldsymbol{x}') \left( \frac{p_\theta(y|\boldsymbol{x}) p_\theta(y|\boldsymbol{x}')}{p_\theta(y)^2} + 1 - 2 \frac{p_\theta(y|\boldsymbol{x})}{p_\theta(y)} \right) \right]^{\frac{1}{2}} \right].
$$

**OvO scenario** The two first terms of the OvO MMD are the same as the first term of the OvA setting, with a simple subscript $a$ or $b$ at the appropriate place. Only the negative term changes. We once again use Bayes' theorem twice:

$$
\mathbb{E}_{\boldsymbol{x}_a \sim p, \boldsymbol{x}_b \sim q}[\kappa(\boldsymbol{x}_a, \boldsymbol{x}_b)] = \mathbb{E}_{\boldsymbol{x}_a \sim p_\theta(\boldsymbol{x}|y_a), \boldsymbol{x}_b p_\theta(\boldsymbol{x}|y_b)} \left[ \kappa(\boldsymbol{x}_a, \boldsymbol{x}_b) \right],
$$
$$
= \mathbb{E}_{\boldsymbol{x}_a, \boldsymbol{x}_b \sim p_{\text{data}}(\boldsymbol{x})} \left[ \frac{p_\theta(y_a|\boldsymbol{x}_a)}{p_\theta(y_a)} \frac{p_\theta(y_b|\boldsymbol{x}_b)}{p_\theta(y_b)} \kappa(\boldsymbol{x}_a, \boldsymbol{x}_b) \right].
$$

The final sum is hence similar to the OvA:

$$
\mathcal{I}_{\text{MMD}}^{\text{ovo}} = \mathbb{E}_{p_\theta(y_a), p_\theta(y_b)} \left[ \text{MMD}(p_\theta(\boldsymbol{x}|y_a), p_\theta(\boldsymbol{x}|y_b)) \right],
$$
$$
= \mathbb{E}_{p_\theta(y_a), p_\theta(y_b)} \left[ \mathbb{E}_{\boldsymbol{x}, \boldsymbol{x}' \sim p_{\text{data}}(\boldsymbol{x})} \left[ \kappa(\boldsymbol{x}, \boldsymbol{x}') \left( \frac{p_\theta(y_a|\boldsymbol{x}) p_\theta(y_a|\boldsymbol{x}')}{p_\theta(y_a)^2} + \frac{p_\theta(y_b|\boldsymbol{x}) p_\theta(y_b|\boldsymbol{x}')}{p_\theta(y_b)^2} \right. \right. \right.
$$
$$
\left. \left. \left. - 2 \frac{p_\theta(y_a|\boldsymbol{x}) p_\theta(y_b|\boldsymbol{x}')}{p_\theta(y_a) p_\theta(y_b)} \right) \right]^{\frac{1}{2}} \right].
$$

### D.3 Wasserstein distance

To compute the Wasserstein distance between the distributions $p_\theta(\boldsymbol{x}|y = k)$, we estimate it using approximate distributions. We replace $p_\theta(\boldsymbol{x}|y = k)$ by a weighted sum of Dirac measures on specific samples $\boldsymbol{x}_i$: $p_N^k$:

$$
p_\theta(\boldsymbol{x}|y = k) \approx \sum_{i=1}^{N} m_i^k \delta_{\boldsymbol{x}_i} = p_N^k,
$$

where $\{m_i^k\}_{i=1}^N$ is the set of weights. We now show that computing the Wasserstein distance between these approximates converges to the correct distance. We first need to show that $p_N^k$ weakly converges to $p$. To that end, let $f$ be any bounded and continuous function. Computing the expectation of such through $p_\theta$ is:

$$
\mathbb{E}_{\boldsymbol{x} \sim p_\theta(\boldsymbol{x}|y=k)}[f(\boldsymbol{x})] = \int_{\mathcal{X}} f(\boldsymbol{x}) p_\theta(\boldsymbol{x}|y = k) d\boldsymbol{x},
$$

which can be estimated using self-normalised importance sampling (Owen, 2013, Chapter 9). The proposal distribution we take for sampling is $p_{\text{data}}(\boldsymbol{x})$. Although we cannot evaluate both $p_\theta(\boldsymbol{x}|y)$ and $p_{\text{data}}(\boldsymbol{x})$ up to a constant, we can evaluate their ratio up to a constant which is sufficient:

$$
\mathbb{E}_{\boldsymbol{x} \sim p_\theta(\boldsymbol{x}|y=k)}[f(\boldsymbol{x})] = \int_{\mathcal{X}} f(\boldsymbol{x}) \frac{p_\theta(\boldsymbol{x}|y = k)}{p_{\text{data}}(\boldsymbol{x})} p_{\text{data}}(\boldsymbol{x}) d\boldsymbol{x},
$$
$$
= \int_{\mathcal{X}} f(\boldsymbol{x}) \frac{p_\theta(y = k|\boldsymbol{x})}{p_\theta(y = k)} p_{\text{data}}(\boldsymbol{x}) d\boldsymbol{x},
$$
$$
\approx \sum_{i=1}^{N} f(\boldsymbol{x}_i) \frac{p_\theta(y = k|\boldsymbol{x} = \boldsymbol{x}_i)}{\sum_{j=1}^{N} p_\theta(y = k|\boldsymbol{x} = \boldsymbol{x}_j)}.
$$

Now, by noticing in the last line that the importance weights are self normalised and add up to 1, we can identify them as the point masses of our previous Dirac approximations:

$$m_i^k = \frac{p_\theta(y = k | \boldsymbol{x} = \boldsymbol{x}_i)}{\sum_{j=1}^N p_\theta(y = k | \boldsymbol{x} = \boldsymbol{x}_j)}.$$

This allows to write that the Monte Carlo estimation through importance sampling of the expectation w.r.t $p_\theta(\boldsymbol{x}|y = k)$ is directly the expectation taken on the discrete approximation $p_N^k$. We can conclude that there is a convergence between the two expectations owing to the law of large numbers:

$$\lim_{N \to +\infty} \mathbb{E}_{\boldsymbol{x} \sim p_N^k}[f(\boldsymbol{x})] = \mathbb{E}_{\boldsymbol{x} \sim p_\theta(\boldsymbol{x}|y=k)}[f(\boldsymbol{x})].$$

Since $f$ is bounded and continuous, the portmanteau theorem (Billingsley, 2013) states that $p_N^k$ weakly converges to $p_\theta(\boldsymbol{x}|y = k)$ when defining the importance weights as the normalised predictions cluster-wise.

To conclude, when two series of measures $p_N$ and $q_N$ weakly converge respectively to $p$ and $q$, so does their Wasserstein distance (Villani, 2009, Corollary 6.9), hence:

$$\lim_{N \to +\infty} \mathcal{W}_c(p_N^{k_1}, p_N^{k_2}) = \mathcal{W}_c\left(p_\theta(\boldsymbol{x}|y = k_1) \| p_\theta(\boldsymbol{x}|y = k_2)\right). \tag{22}$$

# E More information and experiment on the Gaussian and Student-t distributions mixture

## E.1 Generative process of Gaussian and Student Mixture

We describe here the generative protocol for the Gaussian and Student mixture dataset. Each cluster distribution is centered around a mean $\mu_i$ which proximity is controlled by a scalar $\alpha$. For simplicity, all covariance matrices are the identity scaled by a scalar $\sigma$. We define:

$$\mu_1 = [\alpha, \alpha], \quad \mu_2 = [\alpha, -\alpha], \quad \mu_3 = [-\alpha, \alpha], \quad \mu_4 = [-\alpha, -\alpha].$$

To sample from a multivariate Student-t distribution, we first draw samples $x$ from a centered multivariate Gaussian distribution. We then sample another variable $u$ from a $\chi^2$-distribution using the degrees of freedom $\rho$ as parameter. Finally, $x$ is multiplied by $\sqrt{\frac{\rho}{u}}$, yielding samples from the Student-t distribution.

## E.2 Extended experiment

For the main experiment, we fixed $\sigma = 1$, $\alpha = 5$ and the degree of freedom $\rho = 1$. We further tested our method when training a MLP for 4 or 8 clusters. For both cases, we also considered $\rho = 1$ and $\rho = 2$. We include the complete results in Table 6.

# F Model selection on MNIST

We repeat our previous experiment on the MNIST dataset from Sec. 4.4. We choose this time to get 50 clusters at best for both the MI and the MMD GEMINI and train the models for 100 epochs. We repeat the experiment 20 times per model and plot the resulting scores in figures 8a and 8b. We did not choose to test with the Wasserstein GEMINI because its complexity implies a long training time for 50 clusters, as explained in App. G. We first observe in Fig. 8 that the MMD-GEMINI with linear kernel has a tendency to exploit more clusters than the MI. The model converges to approximately 30 clusters in the case of the MLP and 25 for the LeNet-5 model with less variance. We can further observe that for all metrics the choice of architecture impacted the number of non-empty clusters after training. Indeed, by playing a key role in the decision boundary shape, the architecture may limit the number of clusters to be found: the MLP can draw more complex boundaries compared to

Table 6: Mean ARI (std) of a MLP fitting a mixture of 3 Gaussian and 1 Student-t multivariate distributions compared with Gaussian Mixture Models and K-Means. The model can be tasked to find either 4 or 8 clusters at best and the Student-t distribution has $\rho$=1 or 2 degrees of freedom. Bottom line presents the ARI for the maximum a posteriori of an oracle aware of all parameters of the data.

| Model | $\rho = 2$ | | $\rho = 1$ | |
|---|---|---|---|---|
| | 4 clusters | 8 clusters | 4 clusters | 8 clusters |
| K-Means | 0.965 (0) | 0.897 (0.040) | 0 (0) | 0.657 (0.008) |
| GMM (full covariance) | 0.972 (0) | 0.868 (0.042) | 0 (0) | 0.610 (0.117) |
| GMM (diagonal covariance) | **0.973 (0)** | 0.862 (0.048) | 0.024 (0.107) | 0.660 (0.097) |
| $\mathcal{I}_{\mathrm{KL}}^{\mathrm{ova}}$ | 0.883 (0.182) | 0.761 (0.101) | **0.939 (0.006)** | 0.742 (0.092) |
| $\mathcal{I}_{\mathrm{KL}}^{\mathrm{ovo}}$ | 0.731 (0.140) | 0.891 (0.129) | 0.723 (0.114) | 0.755 (0.163) |
| $\mathcal{I}_{\mathrm{H}^2}^{\mathrm{ova}}$ | 0.923 (0.125) | **0.959 (0.043)** | 0.906 (0.103) | 0.86 (0.087) |
| $\mathcal{I}_{\mathrm{H}^2}^{\mathrm{ovo}}$ | 0.926 (0.112) | 0.951 (0.059) | 0.858 (0.143) | 0.887 (0.074) |
| $\mathcal{I}_{\mathrm{TV}}^{\mathrm{ova}}$ | 0.940 (0.097) | 0.973 (0.004) | 0.904 (0.104) | **0.925 (0.103)** |
| $\mathcal{I}_{\mathrm{TV}}^{\mathrm{ovo}}$ | 0.971 (0.005) | 0.620 (0.053) | **0.938 (0.005)** | 0.595 (0.055) |
| $\mathcal{I}_{\mathrm{MMD}}^{\mathrm{ova}}$ | 0.953 (0.060) | 0.940 (0.033) | 0.922 (0.004) | **0.908 (0.016)** |
| $\mathcal{I}_{\mathrm{MMD}}^{\mathrm{ovo}}$ | 0.968 (0.001) | 0.771 (0.071) | 0.921 (0.007) | 0.849 (0.048) |
| $\mathcal{I}_{\mathcal{W}}^{\mathrm{ova}}$ | 0.897 (0.096) | 0.896 (0.021) | 0.915 (0.131) | 0.889 (0.051) |
| $\mathcal{I}_{\mathcal{W}}^{\mathrm{ovo}}$ | 0.970 (0.002) | 0.803 (0.067) | 0.922 (0.006) | 0.817 (0.042) |
| Oracle | 0.991 | | 0.989 | |

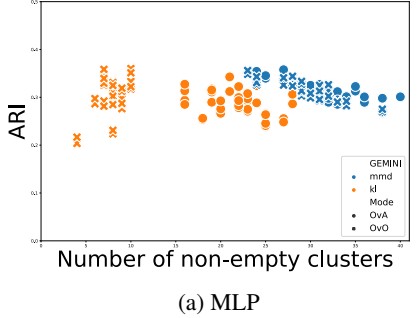

(a) MLP

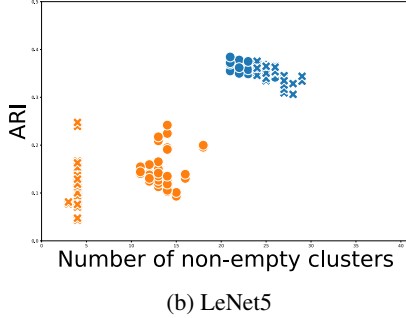

(b) LeNet5

Figure 8: Distributions of the ARI scores given a number of non-empty clusters after 100 epochs of training on MNIST on two different architectures.

the LeNet5 model. Moreover, we suppose that the cluster selection behaviour of GEMINI may be due to optimisation processes. Indeed, we optimise estimators of the GEMINI rather than the exact GEMINI. Finally, Fig. 8 also confirms from Table. 3 the stability of the MMD-GEMINI regarding the ARI despite the change of architecture whereas the MI is affected and shows poor performance with the LeNet-5 architecture.

# G   Choosing a GEMINI

The complexity of GEMINI increases with the distances previously mentionned depending on the number of clusters $K$ and the number of samples per batch $N$. It ranges from $\mathcal{O}(NK)$ for the usual MI to $\mathcal{O}(K^2 N^3 \log N)$ for the Wasserstein-GEMINI-OvO. As an example, we show in Figure 9 the average time of GEMINI as the number of tasked clusters increases for both 10 samples per batch (Figure 9a) and 500 samples (Figure 9b). The batches consists in randomly generated prediction and distances or kernel between randomly generated data.

The Wassertein-OvO is the most complex, and so its usage should remain for 10 clusters or less overall. The second most time-consuming loss is the Wasserstein-OvA, however its tendency in

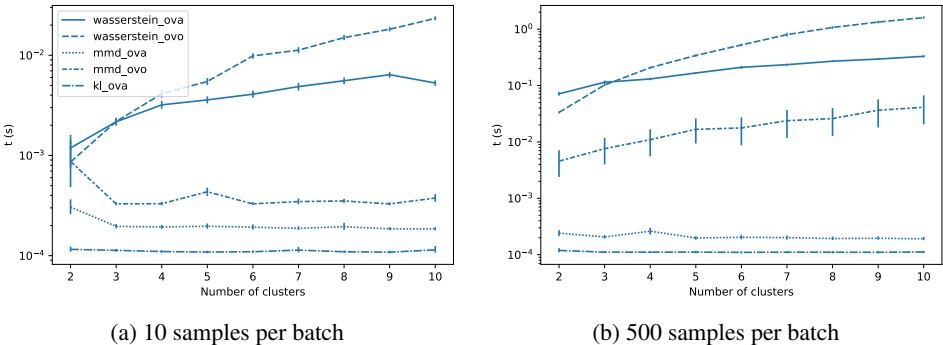

(a) 10 samples per batch          (b) 500 samples per batch

Figure 9: Average performance time (in seconds) of GEMINIs as the number of tasked clusters grows for batches of size 10 and 500 samples.

optimisation to only find 2 clusters makes it a inappropriate. The main difference also to notice between the following MMD is regarding their memory complexity. The MMD-OvA requires only $\mathcal{O}(KN^2)$ while the MMD-OvO requires $\mathcal{O}(K^2N^2)$. This memory complexity should be the major guide to choosing one MMD-GEMINI or the other. Thus, the minimal time-consuming and resource-demanding GEMINI is the MMD-OvA if we consider GEMINIs that incorporates knowledge of data through kernels and distances. Other versions involving $f$-divergences have in fact the same complexity as MI in our implementations, apart from the TV-OvO which reaches $\mathcal{O}(K^2N)$ in our implementation.

## H   All pair shortest paths distance

Sometimes, using distances such as the $\ell_2$ may not capture well the shape of manifolds. To do so, we derive a metric using the all pair shortest paths. Simply put, this metric consists in considering the number of closest neighbors that separates two data samples. To compute it, we first use a sub-metric that we note $d$, say the $\ell_2$ norm. This allows us to compute all distances $d_{ij}$ between every sample $i$ and $j$. From this matrix of sub-distances, we can build a graph adjacency matrix $W$ following the rules:

$$W_{ij} = \left\{ \begin{array}{ll} 1 & d_{ij} \leq \epsilon \\ 0 & d_{ij} > \epsilon \end{array} \right. , \tag{23}$$

where $\epsilon$ is a chosen threshold such that the graph has sparse edges. Our typical choice for $\epsilon$ is the 5% quantile of all $d_{ij}$.

We chose the graph adjacency matrix to be undirected, owing to the symmetry of $d_{ij}$ and unweighted. Indeed, solving the all-pairs shortest paths involves the Floyd-Warshall algorithm (Warshall, 1962; Roy, 1959) which complexity $\mathcal{O}(n^3)$ is not affordable when the number of samples $n$ becomes large. An undirected and unweighted graph leverages performing $n$ times the breadth-first-search algorithm, yielding a total complexity of $\mathcal{O}(n^2 + ne)$ where $e$ is the number of edges. Consequently, setting a good threshold $\epsilon$ controls the complexity of the shortest paths to finds. Our final distance between two nodes $i$ and $j$ is eventually:

$$c_{ij} = \left\{ \begin{array}{ll} \text{Shortest-path}^W(i,j) & \text{if it exists.} \\ n & \text{otherwise} \end{array} \right. . \tag{24}$$

This metric $c$ can then be incorporated inside the Wasserstein-GEMINI.

## I   Packages for experiments

For the implementation details, we use several packages with a python 3.8 version.

- We use PyTorch (Paszke et al., 2019) for all deep learning models and automatic differentiation, as well as NumPy (Harris et al., 2020) for arrays handling.

- We use Python Optimal Transport's function `emd2`(Flamary et al., 2021) to compute the Wasserstein distances between weighted sums of Diracs.

- We used the implementation of SIMCLR from PyTorch Lightning (Falcon et al., 2019).

- Small datasets such as isotropic Gaussian Mixture of score computations are performed using scikit-learn(Pedregosa et al., 2011).

- All figures were generated using Pyplot from matplotlib (Hunter, 2007).