# OpenReview forum: "Generalised Mutual Information for Discriminative Clustering"
_NeurIPS.cc/2022/Conference — NeurIPS 2022 Accept_

### Official Review · Reviewer_usNx · 2022-07-05

**Rating:** 7
**Confidence:** 3
**Soundness:** 4 excellent
**Presentation:** 3 good
**Contribution:** 4 excellent

**Summary:**

The authors introduce measures of discrimenative distance between distributions that are alternatives to mutual information. These measures in the case of MMD and Wasserstein include notions of distance or similarity which can capture the shape of the data distribution. The authors use the proposed measures as objectives for clustering. Evaluation is conducted on supervised multi-class data sets using adjusted rand index as the measure of performance.


**Questions:**

Figure 1: What does "as the posterior turns sharper" mean? This statement is clarified later in the text, but the caption should stand on its own.


**Limitations:**

Nothing to add.

**Strengths And Weaknesses:**

This work deepens our understanding of the clustering problem, an ill-defined and challening problem. By identifying a failure model inherent in using mutual information as an objective, this work opens the possibility to design better clustering algorithms. The proposed methods aim to achieve this through the incorporating of the discrimnative clustering hypothesis: clusters are characterized by a discrimnative model $p(y|x)$.

The MMD and Wasserstein GEMINI include distance information through $k(x,y)$ and $c(x,y)$, respectively. It seems that these regularize the cluster boundaries to be smooth in data space. It would be nice if this intuition could be confirmed/refuted beyond ARI values and supposition. Additionally, while learning the kernel or distance function is (I think) beyond the scope of this paper, it would be good for the authors to discuss what impact the GEMINI approach has on simulaneous kernel learning and clustering methods.

The authors demonstrate some degree of automatic selection of the number of clusters. But, if this is to be a major selling point of the method one would be inclined to set the $k$ to be very large and let the method pick out the correct value. The authors should report performance of such experiments. E.g. how well does the method do on MNIST when $k$ is set to 50 or 100?

Demonstration on more real-world data sets would be welcome.


Typos:
Table 1 MMD-OVA: the x's should be x_a or x_b. Should the y_a and y_b just be y?
Table 2 header: two I_W^OVA.

---

> ### Author Response · Authors · 2022-08-02
> **Re: Official Review of Paper11201 by Reviewer usNx**
>
> Many thanks for your review and understanding of the topic. We corrected Table 1 and Table 2 as suggested.
>
> Regarding the unclarity in Fig. 1, *a model converging to a sharp posterior* means that the distribution $p(y|x)$ converges to the delta Dirac distribution as detailed in appendix A.
>
> > Additionally, while  learning the kernel or distance function is (I think) beyond the scope  of this paper, it would be good for the authors to discuss what impact  the GEMINI approach has on simulaneous kernel learning and clustering  methods.
>
> We think this could be an interesting future work to carry as it could build at the same time meaningful distances/kernels and clusters in some data space. However,  this would question whether the training must remain single-stage by learning simultaneously the kernel/the distance and maximising GEMINI or be two-fold by learning them in an alternating fashion. Moreover, doing so would require a specific attention to initialisation of the models because GEMINIs and distance/kernels are strongly tied.
>
> > The authors should report performance of such experiments. E.g. how well does the method do on MNIST when  is set to 50 or 100?
>
> Thanks for this relevant suggestion: we now provide in the supplementary materials a new experiment on MNIST where we set the limit to 50 clusters for both the MI and the MMD (App. F). We did not experiment with the Wasserstein metric due to its computational complexity. This experiment shows consistent results with Table 3: MMD remains better than KL, both in OvA and OvO. It also shows that the choice of architecture is important, since it allows different types of decision boundaries in the data space that will influence the quantity of clusters. For an MLP, the MMD finds on average 30 clusters. For a LeNet-5, the MMD finds 25 clusters on average.
>
> >  It seems that these  regularize the cluster boundaries to be smooth in data space. It would  be nice if this intuition could be confirmed/refuted beyond ARI values  and supposition
>
> Beyond experiments with ARI validation on supervised data sets, we tried to lead qualitative experiments. With Fig. 2, we highlighted the tendency of MI to converge to inaccurate Dirac distributions in clustering that were solved with MMD-GEMINI. In Fig. 3, we show that the models produced by MMD- or Wasserstein-GEMINI have more conditional entropy around the decision boundary, highliting thus uncertainty even in the presence of a distribution iwht many "outliers" which was overlooked by MI. Finally, figures 4 & 5 illustrate the variation of the decision boundary for GEMINI when we change the core distance as well as using less clusters than tasked.

---

### Official Review · Reviewer_XXcZ · 2022-07-06

**Rating:** 4
**Confidence:** 4
**Soundness:** 2 fair
**Presentation:** 3 good
**Contribution:** 1 poor

**Summary:**

This paper discussed the drawbacks of applying MI and K-L divergence as the distance metric in clustering problems and then proposed some replacements for the K-L divergence. The authors showed that the proposed approach, called generalised mutual information (GEMINI) can outperform the traditional mutual information based approach by several experiments.


**Questions:**

The authors mentioned “We argue that it is intuitive in clustering to compare the distribution of one cluster against the distribution of another cluster rather than the data distribution.” Could the authors explain more about this statement? Why it is more intuitive?

**Limitations:**

No.

**Strengths And Weaknesses:**

This presentation of this paper is clear; however, I am quite skeptical about the contribution of this paper due to the following reasons:

1. The choices of the distance metrics (f-divergence, MMD, Wasserstein) seem to be based on some "intuitive" reasons, without technical justifications. It is unclear to me why a particular choice of a distance metric, say MMD, can outperform another, say Wasserstein, in certain datasets.

2. The justifications of the distance metric designs are mainly based on the experiments and it seems that the authors only conduct limited experiments, and compare with limited approaches. If there are theoretical justification of the proposed approach, I can accept with only limited experiments; however, this paper did not provide such kinds of analyses, and hence I think the experiments are not enough.

3. Overall, I think there is not enough contribution and novelty by just proposing different distance metrics to replace K-L divergence in MI-based clustering. There needs more analyses to justify or prove that which choice of the distance metric can perform well under which kinds of condition or datasets. Without that, it is not convincing for the reviewer that the proposed approach can be either with theoretical guarantees or useful for general classed of datasets.

4. Typo：
line 139 —> “which is possible to estimateSince”

---

> ### Author Response · Authors · 2022-08-02
> **Re: Official Review of Paper11201 by Reviewer XXcZ**
>
> Thanks for your review and questions.
>
> > The authors mentioned “We argue that it is intuitive in clustering to compare the  distribution of one cluster against the distribution of another cluster  rather than the data distribution.” Could the authors explain more about this statement? Why it is more intuitive?
>
> To address the justification of the One-vs-One concept (to compare the distribution of a cluster against another), we provide a new appendix in the supplementary materials (App. B) with a practical example. The necessity of OvO rises when taking into consideration the topology of data through kernels or distances. Indeed, the One-vs-All concept (to compare the distribution of a cluster against the entire data) may miss a cluster when a conditional distribution $p(x|y)$ is similar to $p(x)$ for example when considering expectations. Moreover, the OvA can ensure that two different distributions $p(x|y=i)$ and $p(x|y=j)$ are far from $p(x)$ without necessarily ensuring that both distributions are also far from each other.
>
> > If there are theoretical justification of the proposed approach, I can accept with only limited experiments; however, this paper did not provide such kinds of analyses
>
> We theoretically highlighted with a complete example of Gaussian separation the failure of MI with Fig. 1, Sec. 2.3 and detailed computations in Appendix A. Moreover, to finally show that MI must be improved, we highlight with section 4.1, Fig. 2 that for a categorical model with no information from the data, MI leads to learning a dirac model. This experiment can be viewed as a proxy for deep models. Moreover, we show in section 4.3 and Fig. 4 how our knowledge of the data can lead to the choice of a specific metric for the Wasserstein-GEMINI to be succesful. Finally, even in the absence of knowledge for the best distance regarding the data, we showed consistent results for the novel tool in tables 3&4. We brought further supporting experiment in Appendix E (initial submission), as well as complexity insights on GEMINIs in Appendix F (initial submission). Finally, the proofs of all equations, including the convergence of our Wasserstein estimator are provided in Appendix D, specifically D.3.

---

### Official Review · Reviewer_gtjD · 2022-07-15

**Rating:** 6
**Confidence:** 3
**Soundness:** 3 good
**Presentation:** 2 fair
**Contribution:** 3 good

**Summary:**

This paper aims to solve the problems that arise from using MI in clustering tasks. They generalise the mutual information by changing its
core distance, introducing the generalised mutual information (GEMINI): a set of metrics for unsupervised neural network training.

**Questions:**

Q1. How can we choose which GEMINI method to use?

Q2. What the connections of GEMINI with contrastive learning techniques?

**Limitations:**

Unclear writing:

Line 72: 'Maximising the MI becomes then interesting...'

Line 73: 'However, estimating correctly MI on two continuous domains is hard...'

Line 119: 'Hence regularisation to ensure this intuition is necessary.'

Line 139: 'possible to estimateSince we'

and many more...

**Strengths And Weaknesses:**

Strengths:
1) The paper tackles an important problem.
2) Well motivated
3) The weaknesses of MI are well explained.
4) Good theoretical analysis of GEMINI
5) #clusters need not be known apriori

Weaknesses:
1) Writing needs lot of work
2) Needs more experimental analysis, on more recent, complex datasets
3) Formatting needs work

---

> ### Author Response · Authors · 2022-08-02
> **Re: Official Review of Paper11201 by Reviewer gtjD**
>
> Many thanks for your review and spotted typos.
>
> Q1: Thank you for the concern regarding the choice of GEMINI. We addressed it in the general comment.
>
> Q2: To discuss the link between GEMINI and Contrastive Learning, we first need to clarify the link between MI and CL. As we mentioned in the related works (l. 79-80), contrastive learning losses can be tied to variational lower bounds of MI in case of an intractable distribution $p(x|y)$ (Poole et al, On the variational lower bounds of MI, sec. 2.3).  These lower bounds rely on a key property of MI: adding $N$  random variables $Z_i$ independent from both $X$ and $Y$ does not change the value of MI: $\mathcal{I}(X,Z_1,\cdots,Z_N;Y)=\mathcal{I}(X;Y)$. In the case of CL, these additional random variables are the augmented and adversarial samples. This key property is also verified in GEMINIs OvA and OvO when using $f$-divergences. Therefore, one could potentially create a variational lower bound of $f$-divergence GEMINI for contrastive learning purposes. If we consider IPM-GEMINI like MMD- or Wasserstein-, additional constraints on the kernel or distance must be set to ensure this property. However, CL concerns continous variables X and Y, whereas GEMINIs described in this paper are only for continuous X and discrete Y.

---

### Author Response · Authors · 2022-08-02
**General response from authors**

We would like to thank the reviewers for their valuable feedback and very useful suggestions. We appreciate that "this work deepens our understanding of the clustering problem", has "good theoretical analysis of GEMINI" and that "this presentation [...] is clear".

Our goal was to "tackle an important problem" of Mutual Information by introducing a framework that extends this notion by proposing some "replacements for the K-L divergence", especially those that incorporate information on the data: the Generalised Mutual Information. We then demonstrated how this new notion allows meaningful discriminative clustering, that is to say clustering with any inference model and no prior hypothesis, "an ill-defined and challen[g]ing problem", as noted by all reviewers.

Indeed, the local maxima of mutual information can be irrelevant for clustering purposes as we demonstrated in the example of Section 2.3 and Appendix A, especially when clustering models converge to Dirac distributions in the data space, as highlighted by Fig. 2.a. This implies that the MI is not relevant for discriminative clustering. We further demonstrated how the influence of discriminative distribution design can strongly affect its results in Sections 4.2, 4.4 and 4.5.

Our goal was not to give special insights between data topology and the choice of metrics. We did not intend to create a ready-made solution for clustering.

As rightfully noted by all reviewers, the choice of GEMINI is a key question. All $f$-divergences should be avoided at best; this is one of our main contributions (sec 2.3, Table 3). The problem observed in Figure 2 is extendable to all $f$-divergences. Regarding MMD and Wasserstein, we do not have a clear-defined answer for all cases but rather hints for choosing. We started to discuss it in the supplementary materials: *Choosing a GEMINI* (appendix E in the original submission, G in the new) from a complexity point of view: MMD is faster ($\mathcal{O}(N^2)$ per distance for $N$ samples compared to $\mathcal{O}(N^3)$ for Wasserstein). To provide further insights, we must take into account experimental purposes and context. Indeed, when it is easier to design a distance than a kernel, the Wasserstein GEMINI is more compatible than the MMD GEMINI and vice-versa. Moreover, the MMD-GEMINI inherently computes expectations in a Hilbert Space which allows to compute centroids deemed representative of the clusters. This notion of centroid is less straightforward when using the Wasserstein metric.

To address the concern of all reviewers regarding complex/recent datasets, we tried instead to focus on the clear demonstration of properties highlighted in the presentation of the framework. We showed:

+ the difficulty to converge to Dirac distributions in Sec. 4.1 for GEMINIs,
+ the capability to maintain resistance to outliers like MI in Sec. 4.2 on top of calibrated network,
+ the importance of the distance used with the Wasserstein-GEMINI to improve clustering in Sec. 4.3, including thus a manifold-oriented distance based on neighborhood graphs
+ We showed the similar argument with specific kernels/distances on complex dataset like CIFAR10 in unsupervised learning in Sec. 4.5
+ We finally showed a consistent improvement of MMD-/Wasserstein-GEMINI over $f$-divergence-GEMINI through MNIST

That is why we chose either synthetic datasets that are controlled and well-known deep learning datasets. We did not have unlabeled dataset to put this method in practice  and experts to help us carrying evaluation of clusters. This can be a direction for future works and we are open to suggestions.

Following specific reviewer's comments, we added two new appendices:

1. A geometrical justification and an example of the importance of the One-vs-One concept compared to the One-vs-All (App. B in rebuttal)
2. An additional experiment with 50 tasked clusters on MNIST (App. F in rebuttal)

We thank all reviewers for the suggested corrections. Added appendices and fixes can be found written in red in the rebuttal version.

---

### Meta-Review · Area_Chair_91Pa · 2022-08-25

**Recommendation:** Accept
**Confidence:** Less certain

**Metareview:**

This work presents alternative measures of discriminative distance between distributions that are alternatives to mutual information, addressing the problems that arise from using mutual information in clustering tasks. Drawbacks of applying MI in clustering problems are discussed, and a replacement is proposed by generalizing the mutual information, introducing the generalized mutual information (GEMINI): a set of metrics for unsupervised neural network training. The authors demonstrate that GEMINI can outperform the traditional mutual information-based approach through several experiments.

This is a borderline paper: theoretical justifications and demonstrations on more real-world data sets would improve the paper. However,  the paper addresses an important problem and the results are of interest to the community, and overall the strength overweigh the weaknesses.



**Award:**

No

---

### Decision · Program_Chairs · 2022-09-14

Accept